# Decomposition of phenotypic heterogeneity in autism reveals underlying genetic programs

Aviya Litman [1,2,12], Natalie Sauerwald[3,12] ✉, LeeAnne Green Snyder[4], Jennifer Foss-Feig[4,5,6], Christopher Y. Park[3], Yun Hao[3], Ilan Dinstein[7,8,9], Chandra L. Theesfeld [2,10] & Olga G. Troyanskaya [2,3,10,11] ✉

Unraveling the phenotypic and genetic complexity of autism is extremely challenging yet critical for understanding the biology, inheritance, trajectory and clinical manifestations of the many forms of the condition. Using a generative mixture modeling approach, we leverage broad phenotypic data from a large cohort with matched genetics to identify robust, clinically relevant classes of autism and their patterns of core, associated and co-occurring traits, which we further validate and replicate in an independent cohort. We demonstrate that phenotypic and clinical outcomes correspond to genetic and molecular programs of common, de novo and inherited variation and further characterize distinct pathways disrupted by the sets of mutations in each class. Remarkably, we discover that class-specific differences in the developmental timing of affected genes align with clinical outcome differences. These analyses demonstrate the phenotypic complexity of children with autism, identify genetic programs underlying their heterogeneity, and suggest specific biological dysregulation patterns and mechanistic hypotheses.

At its core, autism spectrum disorder (ASD) is characterized by persistent deficits in social communication and interaction, alongside restricted and repetitive patterns of behavior, interests or activities[1,2]. Numbers of ASD diagnoses have been rising rapidly in recent years, and with the widening diagnostic criteria, there is increasing heterogeneity within the autistic population, phenotypically and genetically[3]. Autism displays a complex phenotypic structure: core features can vary substantially in severity and presentation and can coincide with extensive and unique spectra of associated phenotypes and co-occurring conditions for each individual[2,4]. This wide array of phenotypes is matched by the broad genetic heterogeneity of individuals with autism. Despite substantial evidence for a genetic basis of the condition[5–18] and hundreds of ASD-associated genes having been identified[19–22], we lack a coherent mapping of genetic variation to phenotypes. Here, we leverage broad phenotypic data for a large autism cohort to parse both genetic and phenotypic heterogeneity and identify robust phenotypic classes of individuals and their underlying genetic programs.

Previous studies have demonstrated extensive heterogeneity in autism phenotypes, including cognitive or adaptive behavior[23,24], morphology[25], neuroanatomical imaging profiles[26,27] and clinical

[1]Quantitative and Computational Biology Program, Princeton University, Princeton, NJ, USA. [2]Lewis-Sigler Institute for Integrative Genomics, Princeton University, Princeton, NJ, USA. [3]Center for Computational Biology, Flatiron Institute, New York, NY, USA. [4]Simons Foundation, New York, NY, USA. [5]Department of Psychiatry, Mount Sinai Icahn School of Medicine, New York, NY, USA. [6]Seaver Autism Center for Research and Treatment, Department of Psychiatry, Icahn School of Medicine at Mount Sinai, New York, NY, USA. [7]Cognitive and Brain Sciences Department, Ben Gurion University of the Negev, Be'er Sheva, Israel. [8]Azrieli National Centre for Autism and Neurodevelopment Research, Ben Gurion University of the Negev, Be'er Sheva, Israel. [9]Psychology Department, Ben Gurion University of the Negev, Be'er Sheva, Israel. [10]Princeton Precision Health, Princeton, NJ, USA. [11]Department of Computer Science, Princeton University, Princeton, NJ, USA. [12]These authors contributed equally: Aviya Litman, Natalie Sauerwald. ✉e-mail: nsauerwald@flatironinstitute.org; ogt@genomics.princeton.edu

outcomes[28]. However, linking this heterogeneity to genetic factors has been challenging and limited to trait-centric approaches instead of considering the phenotypes of individuals holistically (that is, the combination of traits of each individual)[12,13,16]. Trait-centric approaches marginalize co-occurring phenotypes when focusing on one trait. However, as traits are not independent, they cannot be separately associated with patterns of genetic variation. During development, traits affect each other in complex ways, compensating for or exacerbating individual phenotype measures. A person-centered approach can capture the sum of these developmental processes at later ages, offering strong clinical value for prognosis with individualized genotype–phenotype relationships. This type of approach has shown promise in applications to other complex psychiatric conditions[29,30].

This person-centered, quantitative phenotypic analysis enables us to address the longstanding challenge of deconvolving the complexity of genetic signals in autism. We leverage a unique cohort with both broad phenotypic and genotypic data at scale ($n = 5,392$) to parse heterogeneity consistent with clinically meaningful presentations of autism[31,32]. Using a generative mixture modeling framework, we decompose phenotypic information to identify, validate and replicate four latent classes, allowing us to associate each of them with different genetic programs. We conduct a thorough investigation of genetic influences in the context of the phenotypic heterogeneity defining our classes, showing that patterns in common genetic variation measured by polygenic scores coincide with their phenotypic and diagnostic traits. We then analyze de novo and rare inherited variation and identify diverging genetic profiles across gene sets and pathways. Finally, we demonstrate that rare variation is associated with class-specific gene expression patterns during development that align with clinical milestones and individual presentations of autism.

## Results

### Identifying the structures of autism phenotypes

To best reflect the complexity of presentations across autistic individuals, we identified 239 item-level and composite phenotype features present in 5,392 individuals from the SPARK cohort[33], a nationwide effort to collect and track genetic and clinical presentations of autism. Briefly, these features represent responses on standard diagnostic questionnaires (the Social Communication Questionnaire-Lifetime (SCQ)[34], Repetitive Behavior Scale-Revised (RBS-R)[35], Child Behavior Checklist 6–18 (CBCL)[36]) and a background history form focused on developmental milestones. These data were analyzed with a general finite mixture model (GFMM) to minimize statistical assumptions while accommodating heterogeneous (continuous, binary and categorical) data types (Methods). The model captures the underlying distributions in the data and provides an inherently person-centered approach, separating individuals into classes rather than fragmenting each individual into separate phenotypic categories (Fig. 1a).

We selected a GFMM with four latent classes representing four different patterns of phenotype profile by considering six standard model fit statistical measures and the overall interpretability of the model solutions. After training models with two to ten latent classes, we found that four classes presented the best balance of model fit as measured by the Bayesian information criterion (BIC), validation log likelihood and other statistical measures of fit (Extended Data Fig. 1 and Supplementary Table 1). In addition, a four-class solution offered the best interpretability in terms of phenotypic separation (Extended Data Fig. 2), as evaluated by clinical collaborators with extensive experience working with autistic individuals. We also found the four-class model to be highly stable and robust to various perturbations (Extended Data Fig. 3).

As observed clinically, classes differed not only in severity of autism symptoms but also in the degree to which co-occurring cognitive, behavioral and psychiatric concerns factored into their presentation. For clinical interpretability, we assigned each of the 239 phenotype features to one of the following seven categories defined in the literature[35,37–39]: limited social communication, restricted and/or repetitive behavior, attention deficit, disruptive behavior, anxiety and/or mood symptoms, developmental delay (DD) and self-injury (Fig. 1b). We identified one class that demonstrated high scores (greater difficulties) across core autism categories of social communication and restricted and/or repetitive behaviors compared to other autistic children, as well as disruptive behavior, attention deficit and anxiety, but no reports of developmental delays; this class was named Social/behavioral ($n = 1,976$). A second class, Mixed ASD with DD ($n = 1,002$), showed a more nuanced presentation, with some features enriched and some depleted among the restricted and/or repetitive behavior, social communication and self-injury categories and overall strong enrichment of developmental delays compared to both nonautistic siblings and individuals in other classes (false discovery rate (FDR) < 0.01; 0.19 < Cohen's $d$ <0.46; Fig. 1c, Extended Data Fig. 4a and Supplementary Table 2). Individuals in the last two classes scored consistently lower (fewer difficulties) and consistently higher than other autistic children across all seven categories. These two classes were termed Moderate challenges ($n = 1,860$) and Broadly affected ($n = 554$). Although individuals in the Moderate challenges class scored below other autistic children across these measured categories, those in all classes still scored significantly higher than nonautistic siblings on the SCQ, the only diagnostic questionnaire with sibling responses, supporting their ASD diagnoses (Fig. 1d). Furthermore, classes displayed significant differences across measures (Supplementary Table 2) and significantly greater between-class variability than within-class variability (Extended Data Fig. 4b), further supporting their phenotypic separation. Additional characteristics of the classes, including sex and age distributions, can be seen in Extended Data Fig. 5.

### Clinical attributes and replication of phenotype classes

The characteristics of the four phenotypic classes we identified were consistent with data on diagnoses of co-occurring conditions and parent reports that were external to our modeling and class identification. A medical history questionnaire, with reports on diagnoses of conditions such as attention-deficit hyperactivity disorder (ADHD), obsessive–compulsive disorder, language delays, depression and anxiety, was not included in the GFMM, but we found that enrichment patterns of these diagnoses matched the class-specific phenotypic profiles and further distinguished the classes phenotypically (Fig. 2a and Supplementary Table 3). The Broadly affected class displayed significant enrichment in almost all measured co-occurring conditions, with the Social/behavioral class matching or exceeding the same diagnostic levels for ADHD, anxiety and major depression (Social/behavioral FDR < 0.01, 1.65 < fold enrichment (FE) < 2.36 compared to out-of-class probands; Fig. 2a), reflecting enrichments in phenotypic profiles (Fig. 1b). The Mixed ASD with DD class was highly enriched in language delay, intellectual disability and motor disorders, compared to both siblings (FDR < 0.01, 8.8 < FE < 20.0) and probands in other classes (FDR < 0.01, 1.38 < FE < 2.33), consistent with the high scores of this class in the categories of developmental delay and restricted and/or repetitive behavior, and individuals in this class showed significantly lower levels of ADHD, anxiety and depression, as expected based on their phenotypic profile. The two classes with greater developmental delays, Mixed ASD with DD and Broadly affected, also showed significantly higher reported levels of cognitive impairment (FDR < 0.01, 1.74 < FE < 3.14), lower levels of language ability (FDR < 0.01, 0.51 < FE < 0.78) and much earlier ages at diagnosis (FDR < 0.01, 0.22 < Cohen's $d$ < 0.98) than the two classes without substantial developmental delays (Fig. 2b, Extended Data Fig. 5d and Supplementary Table 4). In addition, average numbers of interventions (such as medication, counseling, physical therapy or other forms of therapy) were highest among the Broadly affected and Social/behavioral classes (Fig. 2b). These diagnostic data represented the best available external validation, although the natural associations

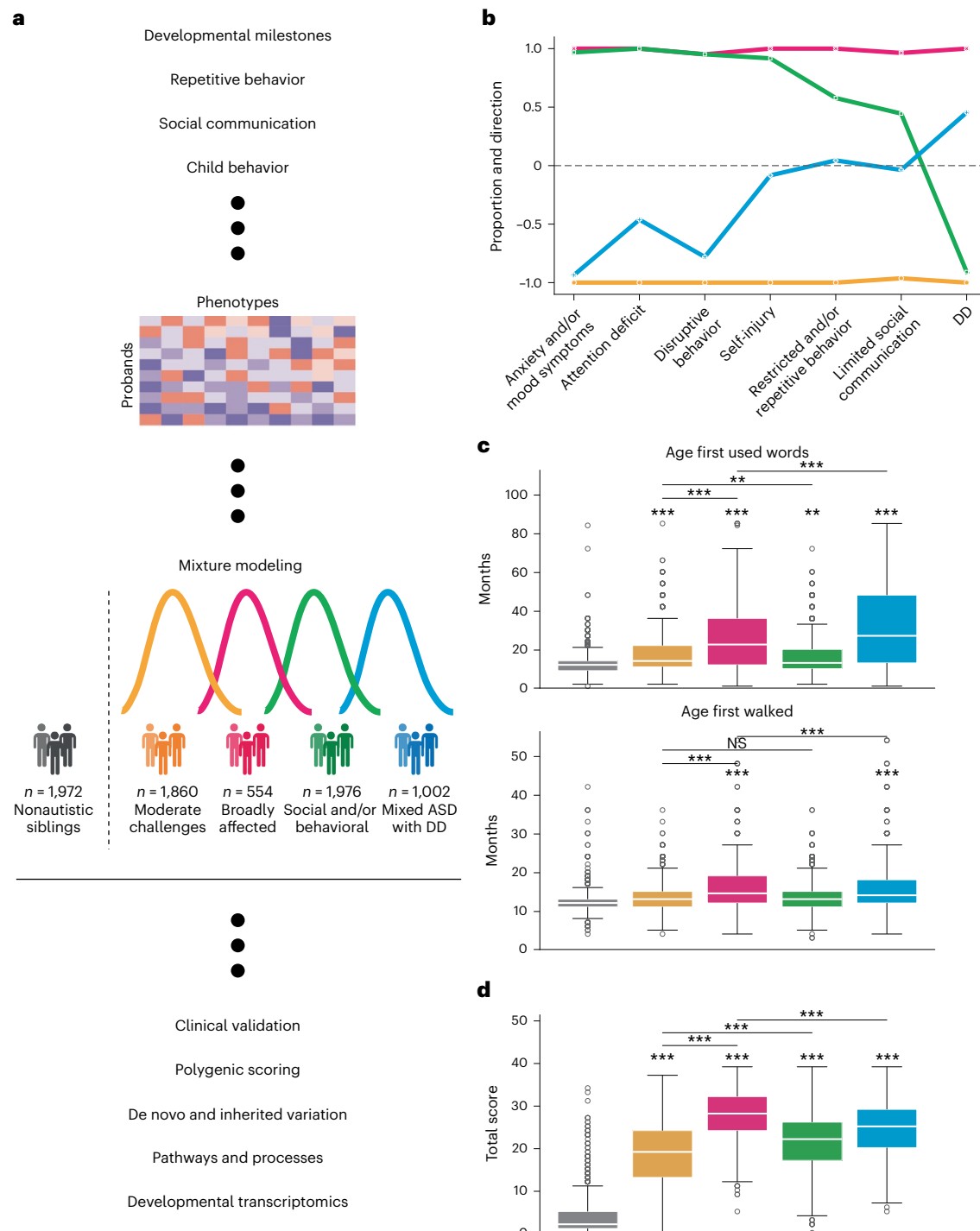

**Fig. 1 | Overview of study design and description of identified subclasses.**
**a**, Study design for parsing the phenotypic heterogeneity of autism and deciphering the genetic factors contributing to individual presentations. A GFMM was trained on a matrix of probands (*n* = 5,392 individuals) by phenotype (239 features describing item-level and composite phenotypic measure data). We describe four data-driven classes of autism that exhibit differing phenotypic presentations and trait patterns. These four subclasses were further characterized by external validations and genetic analyses. **b**, To demonstrate differences in phenotypic patterns, we assessed the propensity of each class toward seven phenotype categories. Values close to 1 indicate that the majority of phenotypes within the category were significantly and positively enriched for the phenotype domain compared to probands in other classes (indicating higher difficulties), and values close to −1 indicate significant negative enrichment or depletion for a given phenotype domain compared to probands in other classes (indicating lower difficulties). Sample sizes for all analyses shown were as follows: Broadly

affected, *n* = 554 (magenta); Social/behavioral, *n* = 1,976 (green); Mixed ASD with DD, *n* = 1,002 (blue); Moderate challenges, *n* = 1,860 (orange); unaffected siblings, *n* = 1,972. **c**, Distributions of two key developmental milestones: the age when the individual first walked and the age when they first used words (both in months) across the four classes, with nonautistic siblings as a control (*n* = 1,972). **d**, Individual total scores from the SCQ by class, with nonautistic siblings as a control (*n* = 1,972). Center lines in all box plots represent the median, box limits represent the 25th and 75th percentiles, whiskers extend to show 1.5× the interquartile range, and outliers are shown separately as open circles. One-sided independent *t*-tests with adjustment for multiple comparisons (Benjamini–Hochberg correction) were used to determine the significance of enrichment for each class compared to siblings, as well as for all possible class–class comparisons (*FDR < 0.1, **FDR < 0.05, ***FDR < 0.01; NS, not significant). Stars directly above the box plot indicate comparisons to siblings, whereas stars accompanied by horizontal bars indicate class–class comparisons. Schematic in **a** created with BioRender.com.

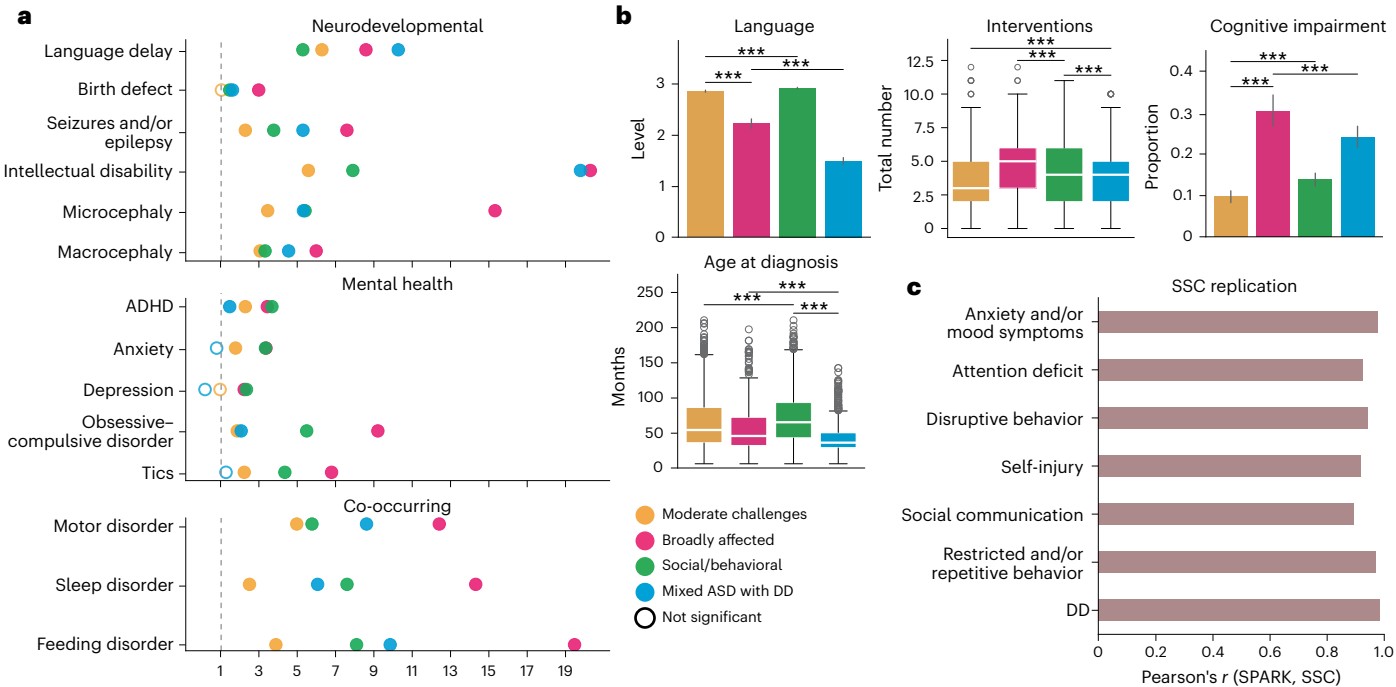

**Fig. 2 | Clinical validation and replication of subclass characteristics.**
**a**, Clinical validation of classes with external medical diagnoses. We computed
the FE (x axis) and statistical significance (FDR) for a selection of available
diagnoses for each class. Open circles indicate FDR > 0.05, whereas closed
circles indicate FDR < 0.05. Statistical comparisons were computed against
siblings as background using one-sided binomial tests and adjusted for
multiple comparisons (Benjamini–Hochberg correction). The dotted line
indicates FE = 1. Sample sizes for all analyses shown were as follows: Broadly
affected, n = 554 (magenta); Social/behavioral, n = 1,970 (green); Mixed ASD
with DD, n = 1,001 (blue); Moderate challenges, n = 1,856 (orange); unaffected
siblings (n = 1,599). **b**, External validation of classes with additional parent-
reported data from background history and medical history questionnaires.
Displayed are: language level at enrollment, parent report with four levels
reflecting language abilities (0, nonverbal; 1, single words; 2, phrases; 3,
sentences), total number of interventions for probands (including options

such as medication, social skills groups, speech therapy and counseling),
cognitive impairment at enrollment, a binary indicator of a diagnosis of
intellectual disability or cognitive impairment, and age at diagnosis in months.
Box plots (center lines represent the median, box limits represent the 25th
and 75th percentiles, whiskers represent 1.5× the interquartile range, open
circles represent outliers) were plotted for continuous variables (one-sided
independent t-tests), whereas bar plots displaying means and 95% confidence
intervals were plotted for binary variables (one-sided binomial tests) and
categorical variables (one-sided independent t-tests; distributions shown in
Extended Data Fig. 5d). *FDR < 0.1, **FDR < 0.05, ***FDR < 0.01. **c**, Replication of
phenotype classes in the SSC. An independent model was trained on the SPARK
dataset (n = 6,393) for features matching across the two cohorts and applied to
the SSC dataset for all individuals with complete data across features (n = 861).
Bars display Pearson correlation coefficients (x axis) between SPARK and SSC
category enrichment proportions across four classes.

between behavioral diagnoses and the behavioral questionnaires on
which our model was trained meant that this was not a fully orthogonal
validation set. However, the consistency observed here further sup-
ported the validity of the self-reported data. Together, these analyses
of medical features show that the four classes were phenotypically
consistent, supporting their separation in genetic analyses.

Furthermore, the four phenotype classes were replicated well in
an independent autism cohort that was deeply phenotyped by trained
clinicians, the Simons Simplex Collection (SSC)[40]. Most phenotypic
questionnaires used in the SPARK model were available for SSC, with the
exception of item-level CBCL data. We combined these matched data,
resulting in 108 training features present for both cohorts. To dem-
onstrate the generalizability of our model to the SSC cohort (n = 861),
we applied a GFMM trained on SPARK data to the SSC test set, as well
as independently training a GFMM on the SSC data. We computed the
enrichment and depletion of each feature within each class across the
seven phenotype categories for both cohorts, as described above for
the original SPARK model. We demonstrated strong replication of the
autism classes in the SSC cohort, with highly similar feature enrichment
patterns across all seven categories (Fig. 2c and Extended Data Fig. 6a).
We further assessed the significance of the overall model similarity with
several permutation tests (Extended Data Fig. 6b,c), shuffling both the
SSC class labels and the SPARK phenotypes before training, and never

observed a higher correlation value with permuted data than the true
correlation of 0.927 ($P < 1 \times 10^{-4}$). The phenotypic classes defined here
are therefore concordant with clinical data and can be replicated in an
external cohort.

**Dissimilar genetic signals underlie phenotypic heterogeneity**
We expected that the differences in phenotypes, co-occurring diagno-
ses and developmental milestones across the four autism classes would
correspond to class-specific patterns in genetic signals for common
variants. We computed polygenic scores (PGS) of children with Euro-
pean ancestry for autism and five other well-powered genome-wide
association studies (GWAS) of related traits and conditions (ancestry
principal component analysis in Extended Data Fig. 7; GWAS refer-
ences in Supplementary Table 5); these showed significant differences
across the four classes that qualitatively matched their clinical and
phenotypic characteristics (Fig. 3a). Several PGS signals within the
classes matched their diagnostic burdens, with the Broadly affected
and Social/behavioral classes showing significantly higher ADHD sig-
nals relative to both nonautistic siblings and other classes (FDR < 0.01,
Cohen's d > 0.22 compared to siblings; FDR < 0.06, Cohen's d > 0.13
for significant cross-class comparisons; Supplementary Table 6) and
significant enrichment of ADHD diagnoses (Supplementary Table 3).
The Social/behavioral class also showed both the highest average PGS

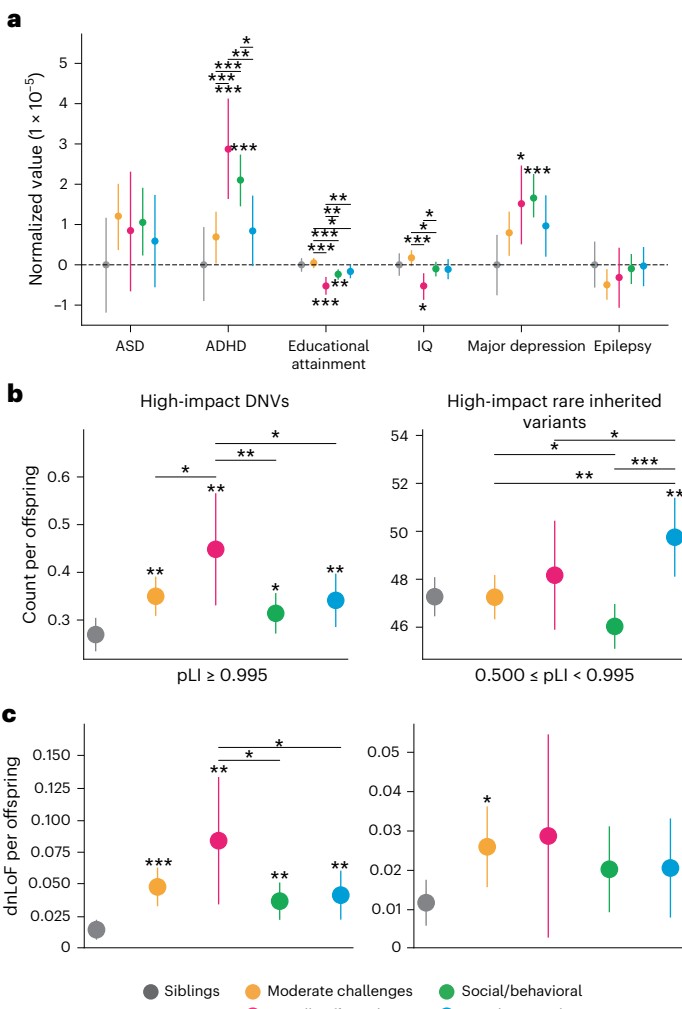

**Fig. 3 | Genetic analyses of genome-wide common and rare variant signals.**
**a**, PGS for ASD GWAS and related phenotypes and conditions. PGS were normalized by the mean of sibling scores within each condition. Sample sizes were as follows: Moderate challenges, *n* = 822; Broadly affected, *n* = 225, Social/behavioral, *n* = 425; Mixed ASD with DD, *n* = 822; unaffected siblings, *n* = 476. **b**, Count per offspring of high-impact DNVs (left) and high-impact rare inherited variants (right) across all protein-coding genes. High-impact variants were defined as variants predicted to be either high-confidence LoF or likely pathogenic missense. Sample sizes for **b** and **c** were as follows: Moderate challenges, *n* = 809; Broadly affected, *n* = 145; Social/behavioral, *n* = 640; Mixed ASD with DD, *n* = 419; unaffected siblings, *n* = 1,013. **c**, Analysis of evolutionarily constrained genes across autism classes and nonautistic siblings. Using the gene-centric measure of evolutionary constraint, pLI, we assigned genes with pLI ≥ 0.5 to one of two categories: pLI ≥ 0.995 (higher constraint genes) or 0.5 ≤ pLI < 0.995 (lower constraint genes). Count burdens (dnLoF) per offspring were then computed for each class. In all parts, circles indicate the mean and error bars show the 95% confidence intervals. Statistical significance was computed with one-sided independent *t*-tests and adjusted for multiple comparisons using Benjamini–Hochberg correction.*FDR < 0.1, **FDR < 0.05, ***FDR < 0.01. Stars above the 95% confidence intervals indicate comparisons to siblings, whereas stars accompanied by horizontal bars indicate direct class comparisons. All statistically significant comparisons are shown.

signal and highest diagnostic burden for major depressive disorder (FDR = 0.00327, Cohen's *d* = 0.204; Figs. 2a and 3a). In addition, the Broadly affected class, which was most enriched for intellectual disability, cognitive impairment and developmental delays, exhibited significantly lower educational attainment and IQ PGS compared with siblings and other classes (FDR < 0.1, Cohen's *d* > 0.17), demonstrating

that co-occurring conditions were associated with common genetic variation that significantly differed among the four identified classes. Notably, none of the classes had a statistically significant signal for the autism PGS, owing to the high variance of this score across our cohort and their siblings.

In addition to dissimilar patterns of common variants, we observed significant differences in rare genetic variation between the phenotypic classes (Fig. 3b and Supplementary Table 7). We conducted de novo and inherited variant calling on the whole exomes of individuals from the cohort using HAT[41] and further classified variants as either loss-of-function (LoF), missense or synonymous. Count burden enrichments were then computed for each variant type in the four classes and among nonautistic siblings (Methods). The Broadly affected class displayed the greatest enrichment for high-confidence de novo LoF (dnLoF) and de novo missense (dnMis) variants compared to both nonautistic siblings and other classes (FDR = 0.01, FE = 1.66 compared to siblings; 0.044 < FDR < 0.086, 1.28 < FE < 1.43 for all class comparisons), whereas the Social/behavioral class displayed the lowest enrichment compared to siblings (FDR = 0.086, FE = 1.17), although we found significant burden in all four classes (Fig. 3b, left; FDR: 0.01, 0.01, 0.086, 0.04). Rare inherited LoF and missense variants displayed statistically significant increases only in the Mixed ASD with DD class (FDR = 0.016, FE = 2.55 compared to siblings), with a significantly higher count per proband in this class compared to the Moderate challenges and Social/behavioral classes (FDR < 0.016, FE > 1.05; Fig. 3b, right; FDR: 0.57, 0.29, 0.97, 0.016). Our analyses differentiated the two classes with greater intellectual disability and developmental delays, showing that the Broadly affected class had more high-impact de novo variants (DNVs), whereas the Mixed ASD with DD class had a combination of high-impact de novo and rare inherited variants compared to nonautistic controls, suggesting a stronger inherited component for the children in this class.

We also observed differences across categories of LoF constraint, including potential significance that had been masked by grouping heterogeneous classes of probands together. Using the gene-level measure of probability of LoF intolerance (pLI)[42], we assigned genes into one of two categories: high-constraint genes (pLI ≥ 0.995) and intermediate-constraint genes (0.5 ≤ pLI < 0.995). When we examined the burden of dnLoF variants across the classes in high-constraint genes, we observed a pattern consistent with our findings above: the Broadly affected class displayed the greatest burden counts in high-constraint genes compared to both siblings and other classes (FDR < 0.1 for two of three classes, 1.76 < FE < 2.3), although there was a significant increase relative to siblings among all autistic classes (Fig. 3c, left; FDR: 6 × 10⁻⁴, 0.013, 0.013, 0.013; odds ratio (OR): 3.69, 6.31, 2.87, 3.25). This finding was consistent with prior work showing an excess burden of mutations in high-constraint genes among probands[9,43]. However, previous work identified no significant increase in intermediate-constraint genes in probands. By separating heterogeneous classes, we found significant enrichment of dnLoF variation in genes of intermediate constraint in the Moderate challenges class (Fig. 3c, right, Supplementary Table 7; FDR: 0.09, 0.26, 0.26, 0.26, OR: 2.54, 2.84, 1.75, 1.95), suggesting that perhaps less-essential genes are affected in this class of individuals. The genetic differences observed between and across classes support the importance of separation of individuals to identify the genetic architectures underlying clinical phenotypic presentations.

## Unique gene sets and pathways associate with phenotypes
We obtained further insight into specific genes and processes dysregulated by the variants across the phenotype classes by investigating the count burdens of de novo and inherited variants in ASD-relevant gene sets[9,19,44–46] (Supplementary Data). Although all four classes displayed enrichment of de novo variation among ASD-related gene sets compared to nonautistic siblings, there were clear differences in the

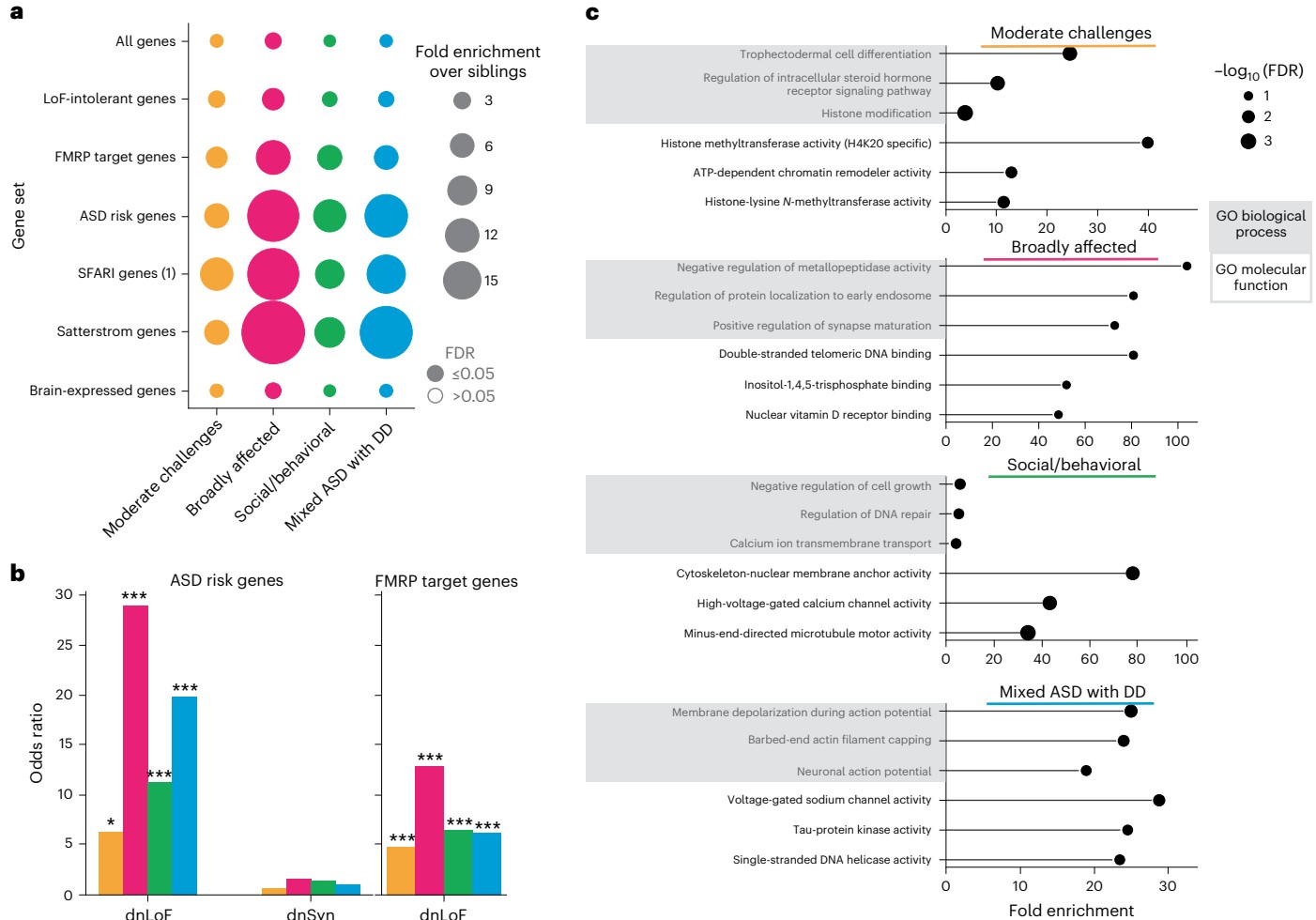

**Fig. 4 | Functional gene set analyses reveal differing genetic profiles.**
**a**, Enrichment and significance of dnLoF burden in each class and gene set. We retrieved seven relevant gene sets and computed the aggregated dnLoF burden for each individual in every gene set. FE (bubble size) and FDR significance (open versus closed circles) were computed relative to nonautistic siblings using one-sided independent *t*-tests followed by adjustment for multiple comparisons (Benjamini–Hochberg correction). An FDR cutoff of 0.05 was used to determine significance. **b**, ORs (*y* axis) across classes for dnLoF variation in ASD risk genes (left) and FMRP target genes (right). ORs for de novo synonymous (dnSyn) variation are displayed for ASD risk genes. Statistical significance was computed with Fisher's exact tests comparing each class to nonautistic siblings and adjusted for multiple comparisons using Benjamini–Hochberg correction. For ASD risk genes, FDR: 0.09, 0.002, 0.009, 0.001 for dnLoF; FDR: 0.85, 0.85, 0.85, 1.0 for dnSyn; for FMRP genes, FDR: 0.004,

0.0004, 0.0005, 0.002 for dnLoF. *FDR < 0.1, **FDR < 0.05; ***FDR < 0.01. **c**, Top significantly affected gene ontology (GO) biological processes and molecular functions are reported for each class against a genome-wide background of protein-coding genes. Gene sets for GO enrichment analyses include all protein-coding genes affected by high-confidence dnLoF or pathogenic missense variation present in individuals from each class. The plots display FE (*x* axis) and log-transformed FDR (bubble size, hypergeometric test with multiple hypothesis correction). Terms were selected by FDR and sorted by FE. For the Moderate challenges, Social/behavioral and Mixed ASD with DD classes, an FDR cutoff of 0.05 was used, whereas a cutoff of 0.1 was used for the Broadly affected class. Shaded boxes represent GO biological processes, and unshaded boxes represent GO molecular functions. Sample sizes in all analyses shown were as follows: Moderate challenges, *n* = 809; Broadly affected, *n* = 145; Social/behavioral, *n* = 640; Mixed ASD with DD, *n* = 419; siblings, *n* = 1,013.

levels and patterns of enrichment among classes (Fig. 4a, Extended Data Fig. 8a,b and Supplementary Tables 8 and 9). For example, all autism-specific gene sets had significantly higher dnLoF mutation burdens (compared to siblings) in the classes with greater developmental delays (average FE: 17.2, 10.9) than in the classes with lower developmental delays (average FE: 5.8, 5.0), suggesting that cognitive outcomes are associated with rare high-impact mutations in a small subset of relevant genes (Fig. 4a). OR analysis further subdivided the classes, with ORs for dnLoF variants being greatest in the classes with greater developmental delays (Fig. 4b, left; FDR: 0.09, 0.002, 0.009, 0.001; OR: 6.3, 28.7, 11.2, 19.7), whereas the ORs for de novo synonymous variants were uniformly distributed, with no significant increases compared with siblings (Fig. 4b, left; FDR: 0.85, 0.85, 0.85, 1.0; OR: 0.7, 1.6, 1.4, 1.1; other gene sets are shown in Extended Data Fig. 8c). Furthermore, DNVs in fragile

X mental retardation protein (FMRP) target genes were strongly associated with the Broadly affected class, with significant enrichment over probands in other classes (FDR = 0.04, FE = 2.2, OR = 12.8). Individuals with fragile X syndrome exhibit developmental delays and intellectual disabilities, as seen across both the Broadly affected and Mixed ASD with DD classes, but they also tend to display mood disorders such as anxiety and impulsive, hyperactive and aggressive behaviors[47], which we observed only in the Broadly affected class.

Molecular pathways affected by the patterns of genetic variation observed across phenotypic classes also suggested different underlying biological mechanisms. Analysis of biological processes affected by high-confidence dnLoF or damaging missense variations in each class revealed little overlap in the top enriched biological processes and no overlap in top molecular functions between all four classes,

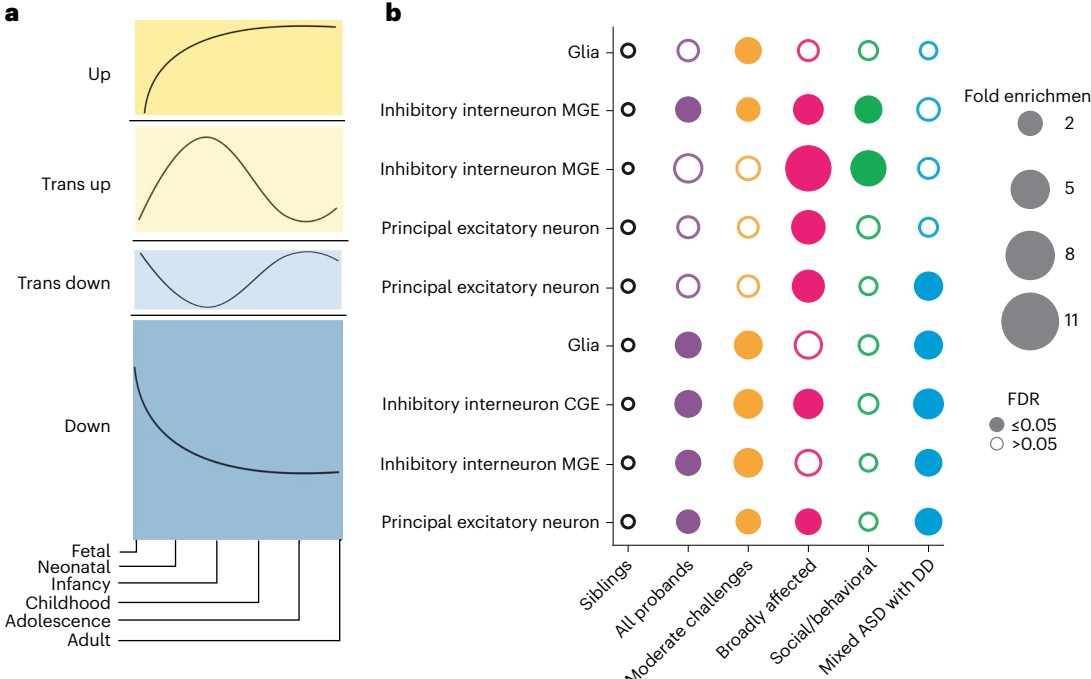

**Fig. 5 | Cell-type-specific and developmental-stage-specific analysis of variant impacts. a**, Trends from Herring et al.[48] representing the gene expression trajectories of brain development genes differentially expressed across developmental stages. Gene expression trajectories follow one of four general patterns: 'up' (first), 'trans up' (second), 'trans down' (third), 'down' (fourth). Trends are measured across the six stages of development (x axis): fetal, neonatal, infancy, childhood, adolescence and adulthood. **b**, Patterns of dnLoF variant enrichment across classes (x axis), major cell types of the prefrontal cortex (y axis) and gene expression trends (y axis). For each class, we computed the FE (bubble size) and corrected P values (FDR, one-sided independent t-tests adjusted for multiple comparisons using Benjamini–Hochberg correction) of

variant burden compared to that of nonautistic siblings. Open circles indicate FDR > 0.05 (not significant), and closed circles indicate significant enrichment (FDR ≤ 0.05). Each column is colored by the corresponding phenotypic class color, with purple representing the combined pool of all probands (n = 2,013). Sample sizes in the analysis were as follows: Moderate challenges, n = 809; Broadly affected, n = 145; Social/behavioral, n = 640; Mixed ASD with DD, n = 419; siblings n = 1,013. Statistics for all class–class comparisons for this analysis can be found in Supplementary Table 11. Cell type and trend combinations with no significant enrichment in any class are not shown. MGE, medial ganglionic eminence; CGE, caudal ganglionic eminence.

suggesting that affected genes represent pathways uniquely associated with class-specific phenotypes (Fig. 4c and Supplementary Table 10). In particular, compared to a gene-based background, the Social/behavioral class was highly enriched for processes of chromatin organization (FE = 3.5, FDR = $1.9 \times 10^{-3}$), regulation of DNA repair (FE = 5.3, FDR = $1.0 \times 10^{-2}$) and microtubule activity (FE = 34.2, FDR = $1.4 \times 10^{-4}$). The Moderate challenges class displayed strong enrichment for histone modification (FE = 3.56, FDR = $1.1 \times 10^{-4}$) and chromatin organization (FE = 3.5, FDR = $2.0 \times 10^{-5}$). By contrast, the Mixed ASD with DD class was characterized by processes of neuronal action potential (FE = 19.0, FDR = $1.0 \times 10^{-2}$) and membrane depolarization (FE = 25.0, FDR = $1.5 \times 10^{-3}$), negative regulation of protein depolymerization (FE = 13.7, FDR = $3.6 \times 10^{-3}$) and voltage-gated sodium channel activity (FE = 28.8, FDR = $3.5 \times 10^{-3}$). Overall, our analysis directly suggests hypotheses for specific biological dysregulations underlying each autism class, providing a framework for directed examination of mechanistic insights in continuing autism research.

## Class-specific developmental gene expression patterns
We found that genes affected by variants in each ASD phenotypic class were associated with unique patterns of gene expression trajectories throughout brain development. This analysis leveraged cell-type-specific developmental gene expression trajectories of the human prefrontal cortex (Fig. 5a and Methods)[48]. We found that the Mixed ASD with DD class was enriched for dnLoF variants that affected genes expressed in all major prefrontal cortex cell types, mostly during the fetal and neonatal stages, with declining expression later in

development (these patterns were termed 'trans down' and 'down' by Herring et al.[48]) (Fig. 5b and Extended Data Fig. 9). By contrast, the Social/behavioral class was enriched only for LoF variants in genes highly expressed postnatally compared to nonautistic siblings (termed 'trans up' and 'up' genes; Fig. 5b) in inhibitory interneurons of the medial ganglionic eminence. Furthermore, the Mixed ASD with DD class was significantly enriched for high-impact variation in genes with the 'down' trend compared to the Social/behavioral class in principal excitatory neurons and inhibitory interneurons (FE > 2, FDR < 0.05; Supplementary Table 11); conversely, the Social/behavioral class was enriched for variation in genes with the 'up' trend compared to the Mixed ASD with DD class (FDR = 0.001, FE = 2.84 for principal excitatory neurons). These developmental gene expression patterns were aligned with the developmental clinical milestones of the classes: the Mixed ASD with DD class had the latest average age of developmental milestone attainment (FDR < $1.9 \times 10^{-19}$, Cohen's d > 0.38 compared to the Social/behavioral class; Fig. 1c and Supplementary Table 2) and the earliest average age of diagnosis (FDR = $6.97 \times 10^{-150}$, Cohen's d = 0.99 compared to the Social/behavioral class; Fig. 2b and Supplementary Table 4). By contrast, the Social/behavioral class, which had variants in later-expressed genes, showed less impact on early development, with later ages at diagnosis and developmental milestones almost in line with those of nonautistic siblings (Cohen's d < 0.07 compared to siblings). The Moderate challenges class displayed enrichment in mostly prenatal gene sets ('down' and 'trans down' genes), although the genes affected in this class tended to be of lower evolutionary constraint than genes affected in the Mixed ASD with DD class, which may have contributed to the differences in

outcomes (Moderate challenges, median pLI = 0.75; Mixed ASD with DD, median pLI = 0.95; $P$ = 0.026). Finally, the Broadly affected class displayed significance for all trends, indicating broad dysregulation across developmental stages and cell types. Our findings thus demonstrate that there are class-specific differences in the developmental timing of genes that are dysregulated, and that these correspond to differences in clinical milestones and outcomes between the classes.

## Discussion

Unraveling the complexity of autism is a particularly challenging yet critical task for supporting the needs of autistic individuals and understanding the biology, inheritance, trajectory and phenotypes of the many forms of the condition. Here, a person-centered approach powered the classification of more than 5,000 autistic children based on a broad set of phenotypes that interact and co-occur, revealing diverging genetic and developmental signals tied to clinical presentations.

The classes we have defined, validated and replicated reflect a robust quantitative analysis of broad phenotypic information in a large sample. The reliance on self-reported data, however, may have introduced rater effects, and intellectual disability is a known confounder in assessing social behaviors. Despite these challenges, we have shown that phenotypic presentation does not reflect a spectrum of intellectual disability, and the self-reported data were both internally consistent with medical history and reproducible in an external cohort with clinician-reported data. The clear separation of genetic signals for each class establishes a concrete set of hypotheses that can be tested with larger cohorts and more comprehensive phenotyping. It will therefore be crucial to expand both the cohort size and the quality and breadth of the phenotyping to more completely capture the full diversity of the autistic population. Coupled with genetic data, future studies incorporating digital phenotypes and longitudinal data will be better equipped to power significant associations between phenotype classes and genotypes, beyond single gene models. Additional genetic information from whole-genome sequencing offers broad potential for discovery of regulatory mechanisms contributing to phenotypic outcomes.

Although all classes had some enrichment for common and rare variants, we found differing signals that defined each autism class. The Social/behavioral class demonstrated the highest signal for common variants associated with co-occurring ADHD and depression, with enrichment for high-impact DNVs in neuronal genes predominantly expressed postnatally. By contrast, the Mixed ASD with DD class was highly enriched for high-impact rare variants, both de novo and inherited, particularly in neuronal genes primarily expressed in utero. The Moderate challenges class was significantly associated with rare variants in genes of lower evolutionary constraint. Finally, the Broadly affected class had the greatest enrichment of high-impact DNVs, specifically in highly constrained genes and FMRP target genes. Some of these trends were consistent with known associations between intellectual disability and genetic pathways, although these classes, which were constructed based on a much richer set of features, provide a more nuanced understanding of autistic individuals and the range of challenges they may face, along with differences in complex genetic architectures that may underlie their condition. By leveraging a person-centered approach, we reveal that these robust, phenotypically separable classes display characteristic patterns of genetic variation. Unlike previous approaches, our analyses directly associate genetic signals with sets of co-occurring phenotypes and further implicate specific affected pathways, brain cell types and developmental stages.

Our findings point to new directions that biologists and neuroscientists could pursue to gain insights into the underlying neurobiology of particular autism presentations and offer the potential for more precise clinical diagnosis and guidance. Future research could also examine how interventions may differ among the classes. In sum, we have demonstrated that person-centered quantitative phenotypic analysis, combined with matched genetic sequencing data at scale, is crucial for uncovering phenotypic classes and the corresponding genetic factors that contribute to the broad heterogeneity of autism.

## Online content

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

## Methods

### Ethics

We received approval to access and analyze deidentified genetic and phenotypic data from the two cohorts from SFARI Base and the Princeton University IRB Committee in the Office of Research Integrity. Our research complies with all relevant ethical regulations.

### Participants

We restricted our sample to children aged 4 to 18 years from the SPARK cohort[33], including both autistic individuals and their nonautistic siblings. The SPARK Phenotype Dataset V9 battery assays core autistic traits (social, language-related and repetitive behaviors), cognitive measures and co-occurring behavioral features (anxiety, aggression and so on). Measures in SPARK with a sufficiently high overlap among participants were selected to maximize the cohort size while maintaining breadth of phenotype data. These measures included the background history (a form that asks parents to report on ages at developmental milestones), SCQ[34], RBS-R[35] and CBCL[36] (Supplementary Note). All autistic children with data from these four phenotype measurement tools were included in the mixture modeling analysis. The same sample of individuals was measured repeatedly. To maximize our sample and feature sizes, we excluded features with less than 90% completeness across the cohort and only included individuals with complete measures across the remaining features. Our resulting cohort included 239 item-level and composite phenotype features and 5,392 individuals in SPARK alone. The cohort had a mean age of 8.56 years with a standard deviation of 3.15 years. Our sibling cohort comprised 1,972 paired siblings (related to at least one proband in the autistic cohort) with no ASD diagnosis, age less than 18 years and available whole-exome sequence data. Of the four phenotypic measures, only the background history and SCQ were available for SPARK siblings. For trio-based analyses, we only used complete trios in SPARK ($n = 1,992$ probands, $n = 837$ paired siblings). In both our phenotypic and genetic analyses, only individuals who passed quality control and had complete data were included.

### Mixture modeling analyses

**Mixture model and covariates.** The phenotype matrix included 5,392 individuals and 241 features (239 training features, two covariates). We used the StepMix package (v.1.2.5)[49] to fit a model and predict labels for each individual using a GFMM approach. This model expands the Gaussian mixture model by using different probability density functions (Gaussian, binomial or multinomial) to model continuous, binary and categorical features. We trained a one-step model with covariates (sex and age at evaluation) and hyperparameters n_init = 200, n_components = 4. The model computed four probabilities per individual to assign each individual to a latent class.

**Stability and robustness analyses.** To demonstrate robustness and stability, we trained 2,000 models with random initializations. We ranked models by their log likelihoods and retained the top 100 models for further analysis. To demonstrate robustness to perturbations of the sample set, we trained 100 independent models (each with 20 random initializations) with a subsampling rate of 50% of the individuals in the sample. In each analysis, we computed the proportion of individuals assigned to the same class, as well as the feature concordance across categories of phenotypes.

**Exploratory class enumeration.** The main tunable hyperparameter of the GFMM is the number of latent classes. It is recommended that this number is chosen by identifying an 'elbow' in statistical measures of model fit, in combination with careful consideration of practical interpretability[50–52]. We tuned this hyperparameter in a multistep process that considered various statistical measures of model fit and indicators of optimality. We evaluated models with 1–12 components and

computed the validation log likelihood, Akaike information criterion, consistent Akaike information criterion, BIC, sample-size-adjusted BIC, likelihood ratio test statistics (Lo–Mendell–Rubin likelihood ratio test) and more, over 50–200 independent runs with randomly generated seeds (Supplementary Note).

**Interpretation.** We consulted clinical collaborators on the interpretability of multiple candidate models and found that the four-class solution offered the best phenotypic separation and most clinically relevant classes. Less complex models offered insufficient phenotypic reduction, whereas more complex models overfitted our data, resulting in fragmented classes that were challenging to interpret owing to mixed enrichments within feature categories. Phenotypic descriptions for three-, five- and six-class model solutions can be found in Extended Data Fig. 2. The final four-class breakdown was as follows: Moderate challenges, $n = 1,860$ (34%); Broadly affected, $n = 554$ (10%); Social/behavioral, $n = 1,976$ (37%); Mixed ASD with DD, $n = 1,002$ (19%). Sex breakdown and percentages by class were as follows: Moderate challenges, 1,459 males (78%), 401 females (22%); Broadly affected, 441 males (80%), 113 females (20%); Social/behavioral, 1,475 males (75%), 501 females (25%); Mixed ASD with DD, 799 males (80%), 203 females (20%).

As outlined above, there was not one clear indicator for this parameter choice. We evaluated a rigorous combination of statistical measures, combined with interpretability, to select our final model: a four-component model that provided the best balance of model fit, complexity and interpretation.

**Phenotype categories.** To phenotypically define and interpret our model outputs, we grouped the 239 features used in training the GFMM into seven factors previously defined in the literature[35,37–39]. Our categories included: limited social communication, restricted and/or repetitive behavior, attention deficit, disruptive behavior, anxiety and/or mood symptoms, developmental delay and self-injury. Each category was a composite of features assigned based on prior grouping evidence.

To more thoroughly examine how the factors defined the four phenotype classes, we performed the following analysis: first, we computed the enrichment (upper-tail significance) or depletion (lower-tail significance) of each feature in each class relative to the other three classes. For binary variables, a one-sided binomial test was used. For categorical and continuous variables, a one-sided independent $t$-test was used. Benjamini–Hochberg multiple hypothesis correction was performed for each autism class and direction of enrichment. We then computed the proportion of features with upper-tail and lower-tail significance in each category and class. At this stage, features were excluded based on three criteria: (1) features that had no significant enrichment or depletion in any class; (2) continuous or categorical features with Cohen's $d$ values between −0.2 and 0.2 across all classes; and (3) binary features with FE of less than 1.5 across all classes. This resulted in 220 contributory features. After computing the proportion of features enriched or depleted in each class and category, we negated the depleted proportions and summed them with the positive enriched proportions. This enabled us to compute a total proportion and direction of significant features for each category and class. This score represented the 'affinity' of the class towards a phenotype category relative to other probands.

**Phenotypic replication in the SSC.** We replicated the four SPARK phenotype classes in an independent cohort, the SSC[40]. We first extracted, processed and integrated the same phenotype measures from SSC that were used in the SPARK model: SCQ, RBS-R, CBCL 6–18 (only composite scores) and a form consisting of parent-reported timing of developmental milestone attainment. We identified 108 phenotype features that were present in both SPARK and SSC. To demonstrate the generalizability of the SPARK model to SSC, we trained a model

on only the SPARK data for the 108 common features with the same hyperparameters as the model described above ($n$ = 6,393 individuals in SPARK with complete data for the 108 features). We compiled the same features for $n$ = 861 individuals in SSC who had complete data across the training features. We then applied the trained model to the SSC test data and predicted a class label for each individual from SSC. At this point, we excluded noncontributory features *post hoc* based on only the training data and according to the criteria described above. We obtained the proportion and direction of enrichment for each class and category across both SPARK and SSC. The Pearson's correlation coefficients between SPARK and SSC for each of the seven phenotype categories were 0.98, 0.92, 0.94, 0.92, 0.89, 0.97 and 0.98.

Next, we obtained a significance of overall model similarity with two permutation tests. First, we randomly shuffled the class labels of SSC participants for $n$ = 10,000 repetitions, obtaining a distribution of chance correlations between SPARK and SSC. We then compared the true correlation between SPARK and SSC (unshuffled labels) to the chance correlation distribution, allowing us to obtain a $P$ value for overall model similarity that accounted for chance correlations. Second, we shuffled the SPARK sample data 1,000 times, each time training an independent model on the shuffled phenotypes, and projected those permuted clusters onto the SSC to test the concordance of the permuted cluster projection. We then compared the distribution of correlations with permuted cluster projections to the correlation with the real cluster projection.

**External phenotypic validation.** Phenotype measures that were available for the majority of our cohort but were not included in model training were used for external validation of the classes and exploratory phenotypic associations of clinical and diagnostic variables. The majority of individuals in our sample cohort ($n$ = 5,381) had a complete or nearly complete basic medical screening form and self-reported (or parent-reported) registration form available. We computed the FE and $P$ values (one-sided binomial test) for each basic medical screening diagnosis within each class compared to nonautistic siblings. For the registration form, we tested for enrichment using one-sided independent $t$-tests for continuous and categorical variables and one-sided binomial tests for binary variables. We also tested all class–class comparisons for every feature. We performed Benjamini–Hochberg correction to correct for multiple comparisons.

### Genetic analyses
**Polygenic scores.** The most recent well-powered GWAS for ASD and several ASD-related traits and co-occurring conditions were used to compute PGS for the subset of our cohort with genotyping array data and European ancestry (determined by a self-reported race of white) passing some basic quality-control metrics ($n$ = 2,294 autistic individuals, $n$ = 426 nonautistic siblings). Genotypes were determined from Infinium Global Screen Array BeadChips available from SPARK[33]. Summary statistics were downloaded and uniformly processed before computing PRS with PLINK's clumping algorithm[53]. Results were regressed by sex and the first six principal components to control for ancestry. All GWAS statistics used were based on populations of European ancestry. Adjustment for multiple comparisons was performed for each GWAS using Benjamini–Hochberg correction.

**DNV calling and filtering.** DNVs were called using the HAT[41] software for all complete trios in our proband and sibling cohort with whole-exome sequence v.3 data ($n$ = 2,013 probands, $n$ = 1,013 paired siblings) from the variant call format files released by the SPARK Consortium. Briefly, this pipeline, which was optimized for whole-genome sequencing and whole-exome sequencing data, used variant calls from both DeepVariant (v.1.1.0)[54] and GATK HaplotypeCaller (v.4.1.2.0)[55] to identify variants that appeared in both variant call sets in the child (genotype of 0/1 or 1/1) but not in either parent (both genotypes 0/0).

Variants were then filtered based on several quality metrics including read depth, genotype quality score and genomic regions. Variants in recent repeats, low complexity regions and centromeres were filtered out. We used the default values for the filtering parameters of minimum depth = 10 and minimum genotype quality score = 20. For filtering and quality control of DNVs, individuals with a high variant count (>3 s.d. above the mean across all trios) were excluded from the analysis, and nonsingleton DNVs (variants appearing in multiple SPARK families) were removed from further processing and analysis. This pipeline identified an average of 2.72 DNVs per proband (s.d. = 2.20) and 2.67 DNVs per sibling (s.d. = 1.63).

**Rare inherited variation.** The DNV pipeline was adapted to identify inherited variants using the same quality metrics. Inherited variants were identified as variants appearing in a child (genotype of 0/1 or 1/1) and in at least one parent (either genotype 0/1 or 1/1). They were then filtered in the same way as the DNVs to identify a high-confidence set of inherited variants. We identified an average of 41,725.2 inherited variants per proband (s.d. = 2,580.7) and 41,778.9 inherited variants per sibling (s.d. = 2,314.6). In all analyses, rare inherited variants were defined as inherited variants with an allele frequency <1% in gnomAD (v.4.1.0).

**Gene sets and resources.** We computed variant enrichments in seven different gene sets, including the set of all protein-coding genes from GENCODE v.29 (Supplementary Data). The SFARI gene set was extracted from SFARIGene and included category 1 genes only[19,22]. The Satterstrom gene set was retrieved from ref. 9. The rest of the gene sets were retrieved from ref. 44 and included genes with pLI > 0.9 from ExAC[56], predicted ASD risk genes (FDR < 0.3) from ref. 6 and target genes of FMRP[45]. Finally, brain-expressed genes were selected using the expression table from GTEx v.7 (gene median transcripts per million per tissue)[46]. The genes selected were those whose expression in brain tissue was at least five times higher than the median expression across all tissues. The high-constraint and moderate-constraint gene sets were selected using pLI scores from gnomAD[42].

**Count-based burden analyses.** For analyses of de novo and inherited variation, we analyzed the distributions of variant count burden for each class of probands or siblings. After extracting de novo and inherited variants for each individual, we ran the variants through Ensembl's Variant Effect Prediction[57] tool (release 111.0) with the – pick flag (most severe consequence) and the LOFTEE[42] (v.1.0.4) and AlphaMissense[58] plugins. To predict LoF variants, we flagged variants with the following predicted consequences: stop gained, frameshift variant, splice acceptor variant, splice donor variant, start lost, stop lost and transcript ablation. We further subset the variants to only include those flagged as high-confidence ('HC') by LOFTEE. To predict high-confidence missense mutations, we flagged variants with the following predicted consequences: missense variant, inframe deletion, inframe insertion and protein-altering variant. We further subset the variants to those predicted as 'likely pathogenic' by AlphaMissense. Some individuals had zero exome-wide calls for DNVs ($n$ = 91 probands and $n$ = 67 siblings). We accounted for their variant counts in each count-based analysis of DNVs.

We aggregated the counts of dnLoF, dnMis, rare inherited LoF variants and rare inherited missense variants for each individual. Modeling the variant counts as a distribution for each class and siblings, we performed hypothesis testing (one-tailed independent $t$-tests) followed by Benjamini–Hochberg correction for each variant class. Hypothesis testing was performed for all class–sibling and class–class comparisons.

**Gene set and OR analyses.** We computed the gene-set-specific variant burden counts and repeated the same hypothesis testing procedure using one-sided independent $t$-tests followed by Benjamini–Hochberg

correction. All proband variant count distributions were tested against the sibling variant count distribution, and each class was also tested relative to all out-of-class probands and other classes. ORs were computed for each phenotype class and gene set. All comparisons were made for each class against siblings using Fisher's exact tests followed by Benjamini–Hochberg correction for each gene set and variant type separately.

**GO term analysis.** To extract the top disrupted biological processes for each class, we flagged dnLoF and dnMis variants present in all protein-coding genes across individuals in each class. To extract the top GO biological processes and molecular functions for each class, we used ShinyGO 0.80 (ref. 59) (results generated on September 27, 2024), which computes FE and FDR enrichment values using a hypergeometric test, with all protein-coding genes as background. We selected the terms by FDR enrichment and sorted for top terms by FE values. For the Moderate challenges, Social/behavioral and Mixed ASD with DD classes, we used an FDR cutoff of 0.05. For the Broadly affected class, we used an FDR cutoff of 0.1 owing to the smaller class size.

**Developmental gene expression analysis.** We leveraged a single-cell human prefrontal cortex dataset collected from postmortem tissues at six different stages of development[48]. We used Table S3 from ref. 48 to retrieve brain development gene sets (termed devDEGs) for prefrontal cortex cell types that followed one of four general gene trend patterns defined by the authors: 'up' (increasing expression throughout stages, with the highest expression in late stages), 'trans up' (increasing expression in earlier stages, peak in middle stages, followed by a decline in later stages), 'down' (declining expression throughout stages, with the highest expression in the fetal stage) and 'trans down' (declining expression in early stages, lowest expression in middle stages and increasing expression in late stages). devDEGs were identified through a differential expression analysis of the six developmental stages (fetal, neonatal, infancy, childhood, adolescence, adulthood). Although 14 major trajectories were identified, we opted to use the four general gene trends for clarity and interpretability. We further combined clusters of cells into the four identified major cell type categories: principal excitatory neurons, inhibitory interneurons (medial ganglionic eminence), inhibitory interneurons (caudal ganglionic eminence) and glia. This was done by taking the union of devDEGs associated with each major cell-type category. We conducted a count burden analysis of dnLoF mutations in each phenotype class compared to nonautistic siblings and for every class–class comparison and computed FE values (of the means) and P values (one-sided independent t-tests) followed by Benjamini–Hochberg correction.

To compare the constraint of genes affected in the Moderate challenges and Mixed ASD with DD classes, we extracted the genes with high-confidence dnLoF variants in each class for the following combinations of gene expression trends and cell types: down in principal excitatory neurons, down in inhibitory interneurons (medial ganglionic eminence), down in inhibitory interneurons (caudal ganglionic eminence) and down in glia. These categories displayed enrichment for dnLoF variants across both classes. We extracted the pLI values for affected genes in each class and conducted a one-tailed Mood's median test to determine whether the distributions of pLI scores were significantly different between the two classes.

### Reporting summary

Further information on research design is available in the Nature Portfolio Reporting Summary linked to this article.

### Data availability

To abide by the informed consent that individuals with autism and their family members signed when agreeing to participate in a SFARI cohort (SSC and SPARK), researchers must be approved by SFARI Base (https://base.sfari.org).

### Code availability

Code and scripts used for the analyses presented in this manuscript are available via GitHub at https://github.com/Function-Lab/asd-pheno-classes and via Zenodo at https://doi.org/10.5281/zenodo.15324658 (ref. 60).

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

### Acknowledgements

We thank all the families in SPARK, the SPARK clinical sites and SPARK staff. We acknowledge access to the SPARK phenotypic and genetic datasets on SFARI Base. Approved researchers can obtain the SPARK population dataset described in this study by applying at https://base.sfari.org. This work used computing resources supported by the Scientific Computing Core at the Flatiron Institute and was supported by funding from the NIH NIGMS no. R01GM071966 (O.G.T.), Simons Foundation grant 395506 (O.G.T.) and NIH NHGRI training grant T32HG003284 (A.L.).

### Author contributions

A.L. and N.S. contributed equally to this work, conceiving the study, performing analyses, and writing the paper with the help of C.L.T. and O.G.T. L.G.S. and J.F.-F. provided clinical expertise and supervision for phenotype analyses. C.Y.P., Y.H. and I.D. proposed experiments and gave advice on genetic analyses.

### Competing interests

The authors declare no competing interests.

### Additional information

**Extended data** is available for this paper at https://doi.org/10.1038/s41588-025-02224-z.

**Correspondence and requests for materials** should be addressed to Natalie Sauerwald or Olga G. Troyanskaya.

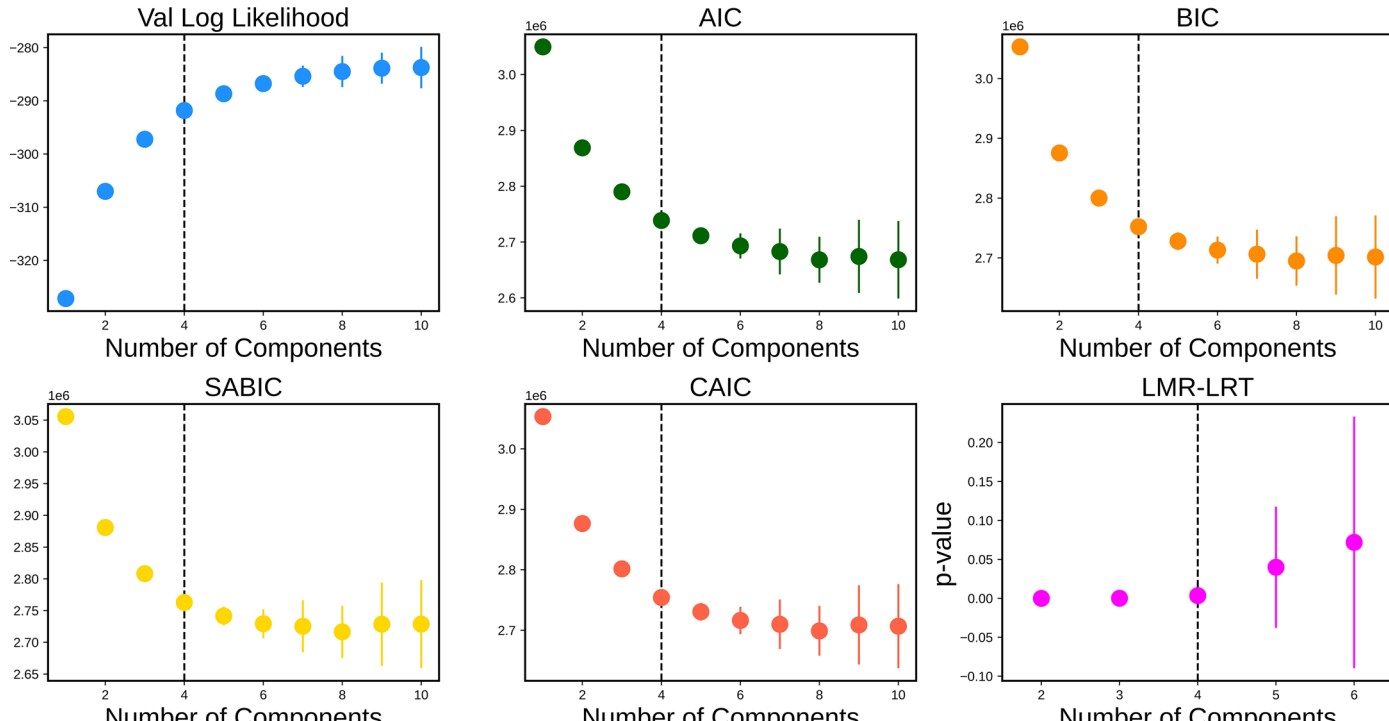

**Extended Data Fig. 1 | Exploratory class enumeration analyses suggest four components for the SPARK generative mixture model.** A crucial step in mixture modeling is tuning the number of model components through evaluation of fit metrics. Models were trained with varying numbers of components (*x*-axis) on the phenotype data of *n* = 5,392 probands over 50–200 iterations with randomly generated seeds. Points represent means and error bars represent the standard deviation for a variety of statistical indicators (*y*-axis): Validation Log Likelihood (LL), Akaike Information Criterion (AIC), Bayesian Information Criterion (BIC), Sample-Size-Adjusted BIC (SABIC), Consistent AIC (CAIC), and the Likelihood Ratio Test (LRT). The evaluation of these indicators combined with interpretation of each candidate model and class resulted in the selection of a four-component model (shown as vertical dotted line in all plots).

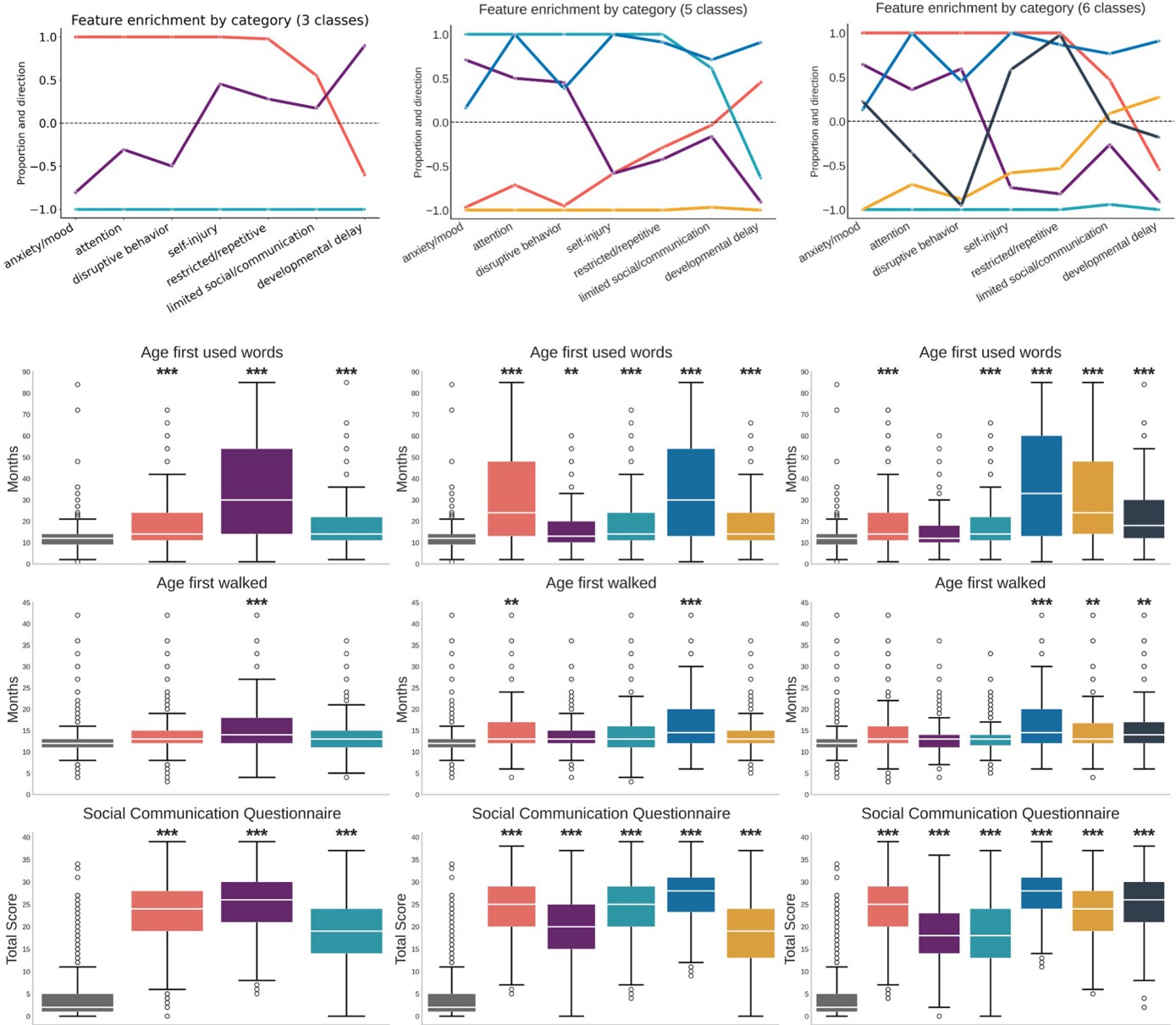

**Extended Data Fig. 2 | Model characteristics for 3-, 5-, and 6-class models.** In addition to a wide range of statistics, we also considered interpretability in selecting the four-class model. Top plots represent proportion and direction of affinity of each class (*y*-axis) towards each of seven phenotype categories (*x*-axis). Bottom plots show SCQ total score and developmental milestone scores (*y*-axis) for proband classes (*n* = 5,392 total) and unaffected siblings (*n* = 1,972). Though the three-class model (*n* = 1,868, 1,068, and 2,456) also shows separation between classes, there remains a lot of within-class variation, suggesting that additional classes would better capture the heterogeneity of the cohort. The five-class model (*n* = 781, 1,828, 949, 462, and 1,372) is challenging to interpret, with mixed enrichment patterns and overlap between classes. The six class

solution (*n* = 929, 1,356, 1,263, 390, 690, and 764) is even less interpretable, likely overfitting to our cohort and splitting classes with substantial similarities. All models were trained on phenotype data from *n* = 5,392 probands in the sample. Center lines in all boxplots represent the median, box limits represent the 25th and 75th percentiles, whiskers extend to show 1.5× interquartile range, and outliers are shown separately as open circles. One-sided independent *t*-tests with adjustment for multiple comparisons (Benjamini-Hochberg correction) were used to determine significance of enrichment for each class compared to siblings (* indicates FDR < 0.1, ** indicates FDR < 0.05, and *** indicates FDR < 0.01 in all figures). Stars directly above the boxplot indicate comparisons to siblings.

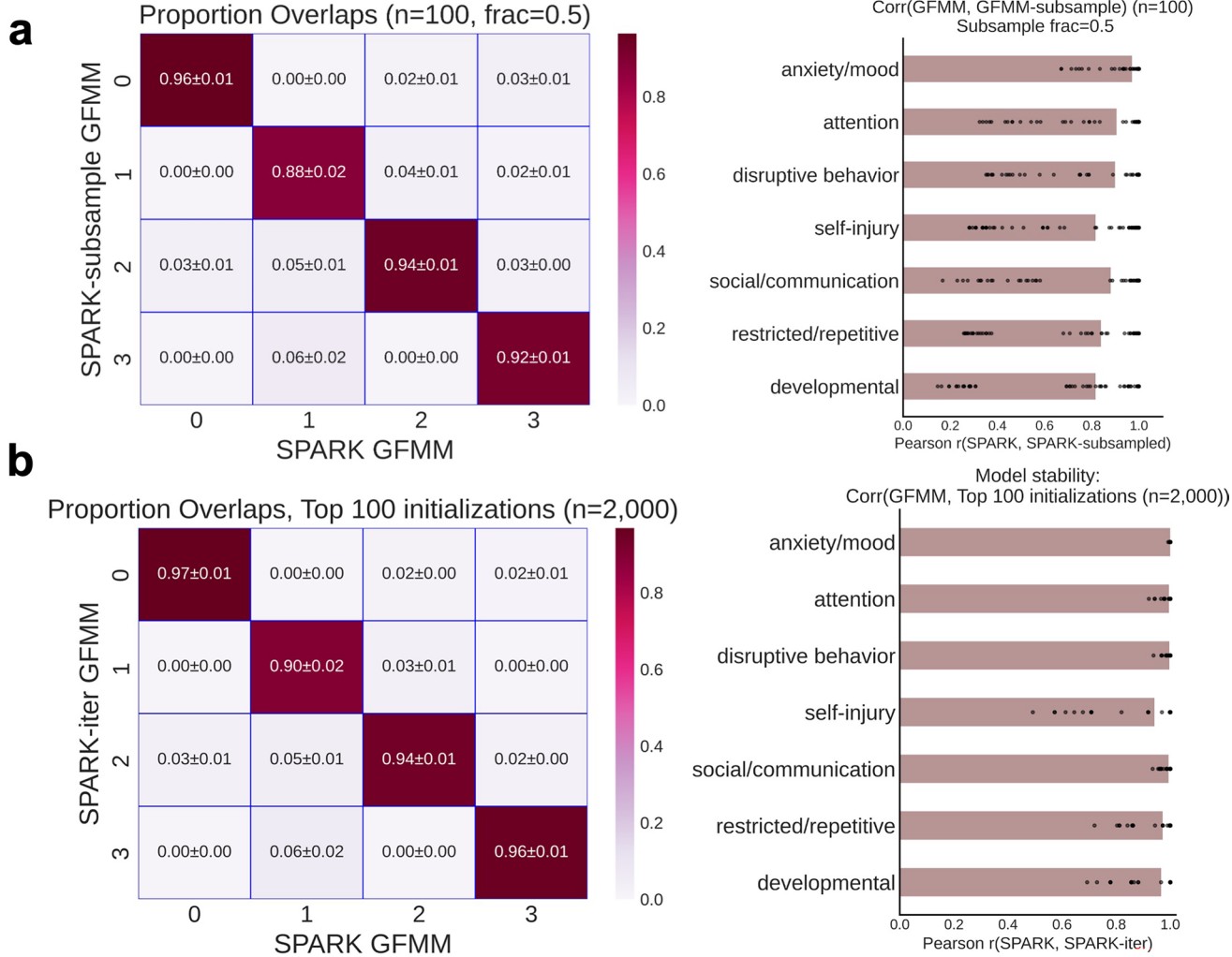

**Extended Data Fig. 3 | Model stability and robustness to subsampling and random initializations. a**, We trained 100 independent models (each with 20 random initializations) with a subsampling rate of 50% of individuals in the sample ($n = 2,696$). We compared both the individual class assignments and the overall phenotypic correlations between the subsampled models and the full model. We observed very strong stability across the subsampled models, with averages of 88–96% of individuals assigned to the same group as the full model, as shown in the heatmap representing the proportion of individuals assigned to the same group in the subsampled model as the full model. Additionally, the feature concordance among subsamples and across categories of phenotypes remains very stable, as shown in the bar plot quantifying correlations between the feature enrichments of the subsampled model and the full model (x-axis). In all displayed plots, values and bars represent means, error intervals represent the 95% confidence interval, the color bar varies continuously from low proportion

(white) to high proportion (magenta), and individual data points are overlaid on the right. **b**, We trained 2,000 independent models on the full sample ($n = 5,392$) with random initializations, each time recording the log likelihood (LL) of the model. We ranked the trained models by their LL and retained the top 100 models for further analysis. We observe very strong stability across the top randomly initialized models, with averages of 90–97% of individuals assigned to the same group as the original model reported in the manuscript, as shown in the heatmap representing the proportion of individuals assigned to the same group in the randomly initialized models as the original model. Additionally, the feature concordance among randomly initialized models and across categories of phenotypes remains very stable, as shown in the bar plot quantifying correlations between the feature enrichments of the randomly initialized models and the original model (x-axis).

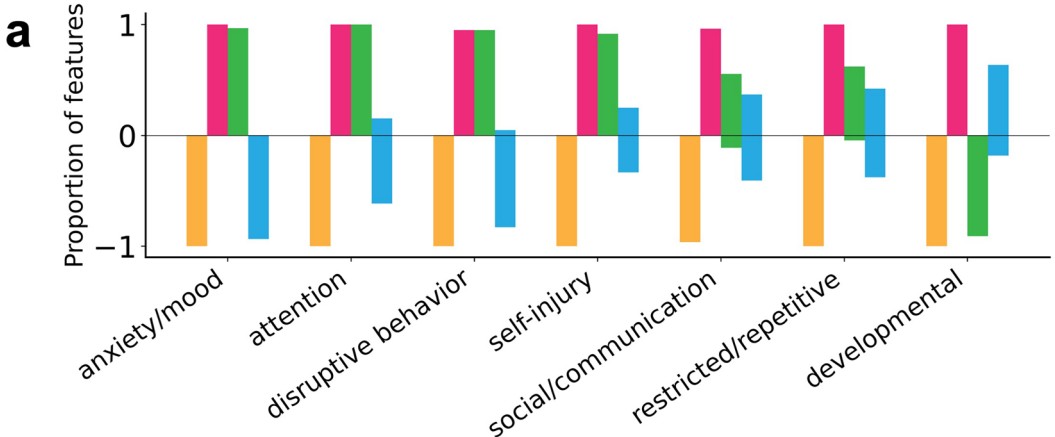

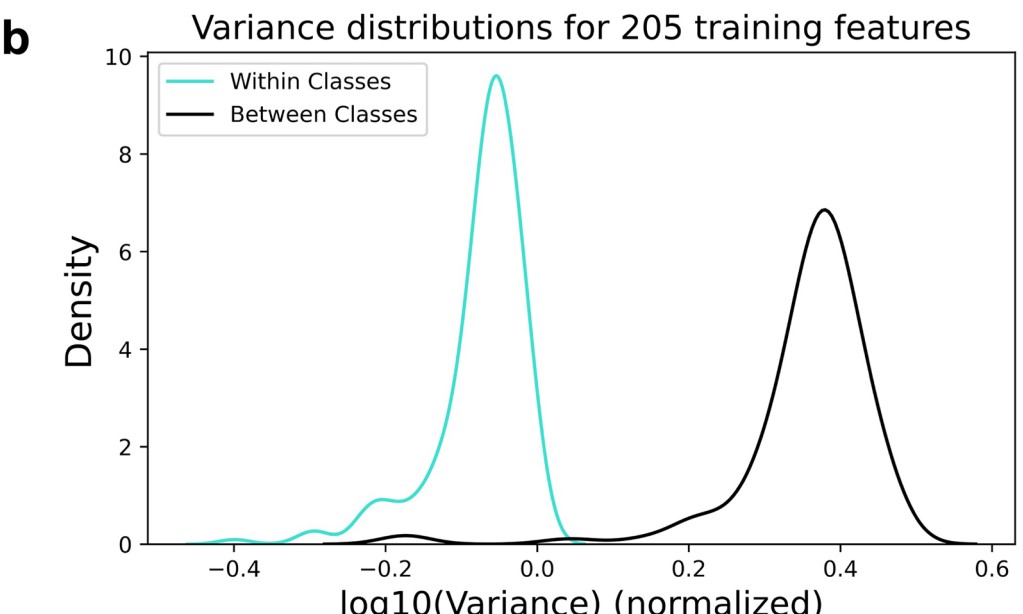

**Extended Data Fig. 4 | Comparison of feature variances. a**, Variability of feature enrichment and direction of enrichment computed for the sample ($n = 5,392$). To demonstrate variability in feature significance within phenotype categories and classes, enrichments and depletions are plotted for each combination of phenotype category and class, with the $y$-axis representing the proportion and direction of enrichment of features assigned to the phenotype category ($x$-axis) and computed within the class. Sample sizes for analyses shown are: Broadly affected (magenta, $n = 554$), Social/behavioral (green, $n = 1,976$), Mixed ASD with DD (blue, $n = 1,002$), Moderate challenges (orange, $n = 1,860$), and unaffected siblings ($n = 1,972$). **b**, We computed the normalized variance ($x$-axis) of each feature across and within classes for $n = 5,392$ probands. To obtain the distribution of within class variances, we computed the weighted average of the variances across the four classes for each feature. We plotted the distributions of feature variances, demonstrating that variances between classes significantly exceed variances within classes for $n = 205$ features ($P = 9.97 \times 10^{-176}$; one-sided independent $t$-test).

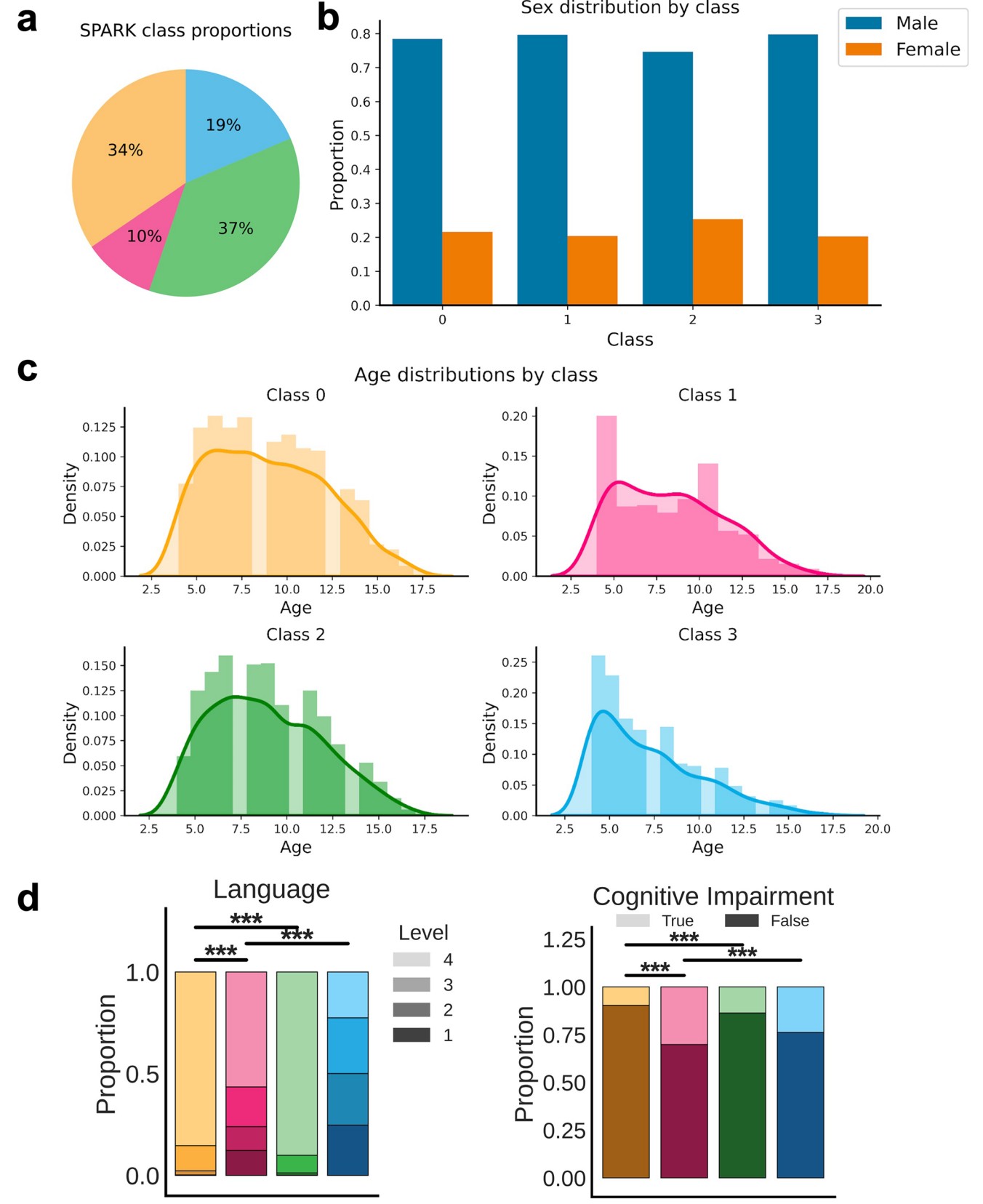

**Extended Data Fig. 5 | See next page for caption.**

**Extended Data Fig. 5 | Age and sex distribution breakdown by class.**
**a**, Breakdown of proportion of SPARK cohort individuals (total $n = 5,392$) assigned by the generative mixture model to each latent class. Classes are represented by the corresponding class colors and numbers are rounded to the nearest percent. **b**, Proportion of males and females ($y$-axis) for individuals (total $n = 5,392$) assigned to each latent class. **c**, Density of age distribution ($x$-axis) for individuals assigned to each latent class (total $n = 5,392$). Colors correspond to class colors. **d**, Stacked bar plots showing distribution of proportions ($y$-axis) across classes ($x$-axis) for two variables: language level at enrollment, a parent report with four levels reflecting language abilities

(0 = Nonverbal, 1 = Single words, 2 = Phrases, 3 = Sentences), and cognitive impairment at enrollment (binary True/False). Sample sizes are: Broadly affected (magenta, $n = 554$), Social/behavioral (green, $n = 1,976$), Mixed ASD with DD (blue, $n = 1,002$), and Moderate challenges (orange, $n = 1,860$). One-sided binomial tests (cognitive impairment) and one-sided independent $t$-tests (language level) were used to test for significance between classes (data not available for unaffected sibling control). $P$-values were adjusted for multiple comparisons using Benjamini-Hochberg correction. * indicates FDR < 0.1, ** indicates FDR < 0.05, and *** indicates FDR < 0.01 in all figures.

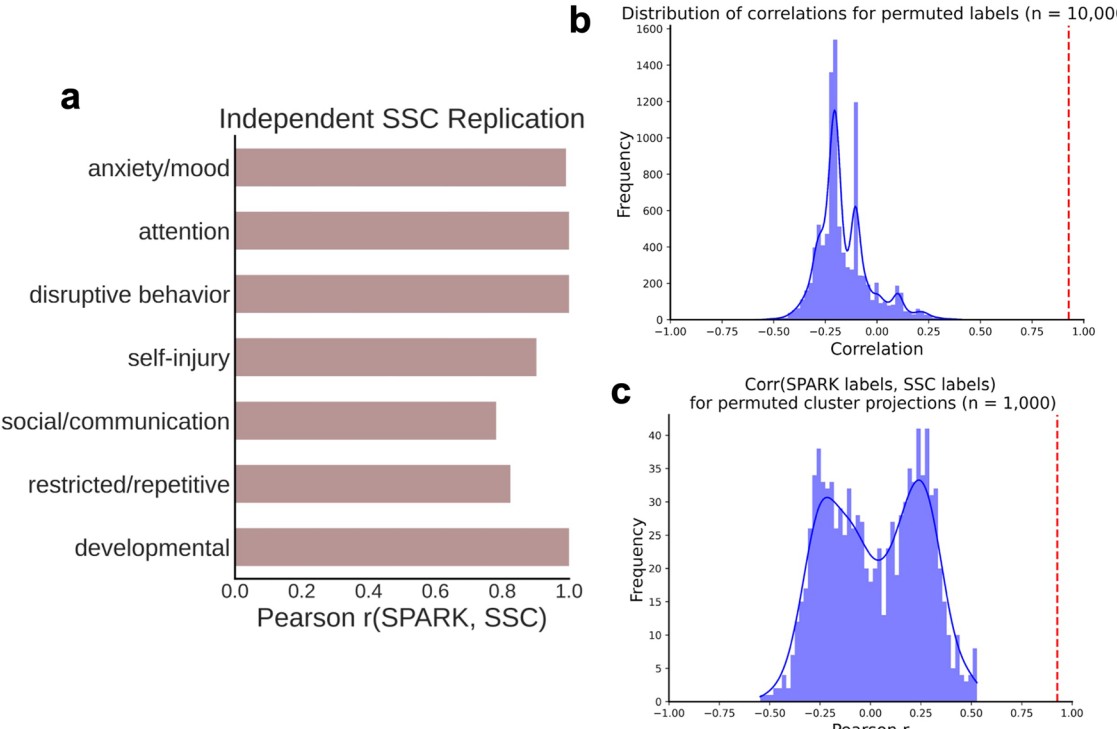

**Extended Data Fig. 6 | Phenotypic replication of autism classes in the Simons Simplex Collection (SSC). a**, Independent phenotypic replication of classes in the SSC cohort. A model was independently trained on the Simons Simplex Collection (SSC) dataset for 108 matching features and $n$ = 861 probands. The feature enrichment patterns were computed for the seven phenotype categories ($y$-axis) and Pearson correlation coefficients with the full SPARK model ($n$ = 5,392) are shown ($x$-axis). **b**, Permutation analysis of SSC phenotypic replication. Pearson correlation coefficients ($x$-axis) computed between the feature enrichment patterns of latent classes in SSC and SPARK for 10,000 random permutations of class labels in the SSC. The dashed vertical red line represents the true correlation from applying the model trained on SPARK data to predict labels on individuals in SSC (Pearson's $r$ = 0.927). No chance

correlations exceeded the true correlation. Model was trained on $n$ = 6,393 probands in SPARK and applied to $n$ = 861 probands in the SSC. **c**, Pearson correlation coefficients ($x$-axis) computed between the feature enrichment patterns of latent classes in SSC and SPARK for 1,000 permuted cluster projections onto the SSC. All models were trained independently on the shuffled phenotype labels of the SPARK dataset (shuffling was repeated for every model). The permuted phenotype models were projected onto the SSC and correlated with the true SPARK labels. The dashed vertical red line represents the true correlation from applying the model trained on SPARK data to predict labels on individuals in SSC (Pearson's $r$ = 0.927). No chance correlations exceeded the true correlation. Models were trained on $n$ = 5,392 probands in SPARK and $n$ = 861 probands in the SSC.

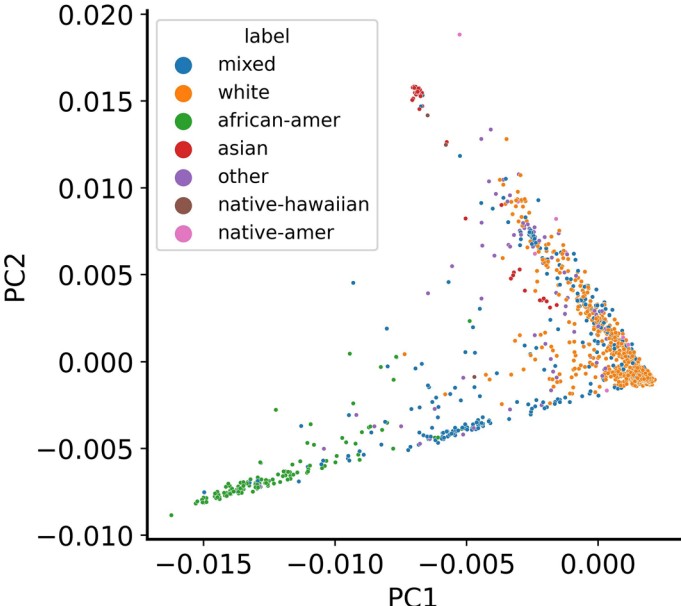

**Extended Data Fig. 7 | Ancestry PCA plot for genotyped cohort.** Top two principal components of the SNP data (*x*-axis, *y*-axis) representing genetic ancestry. Points are colored by self-reported race.

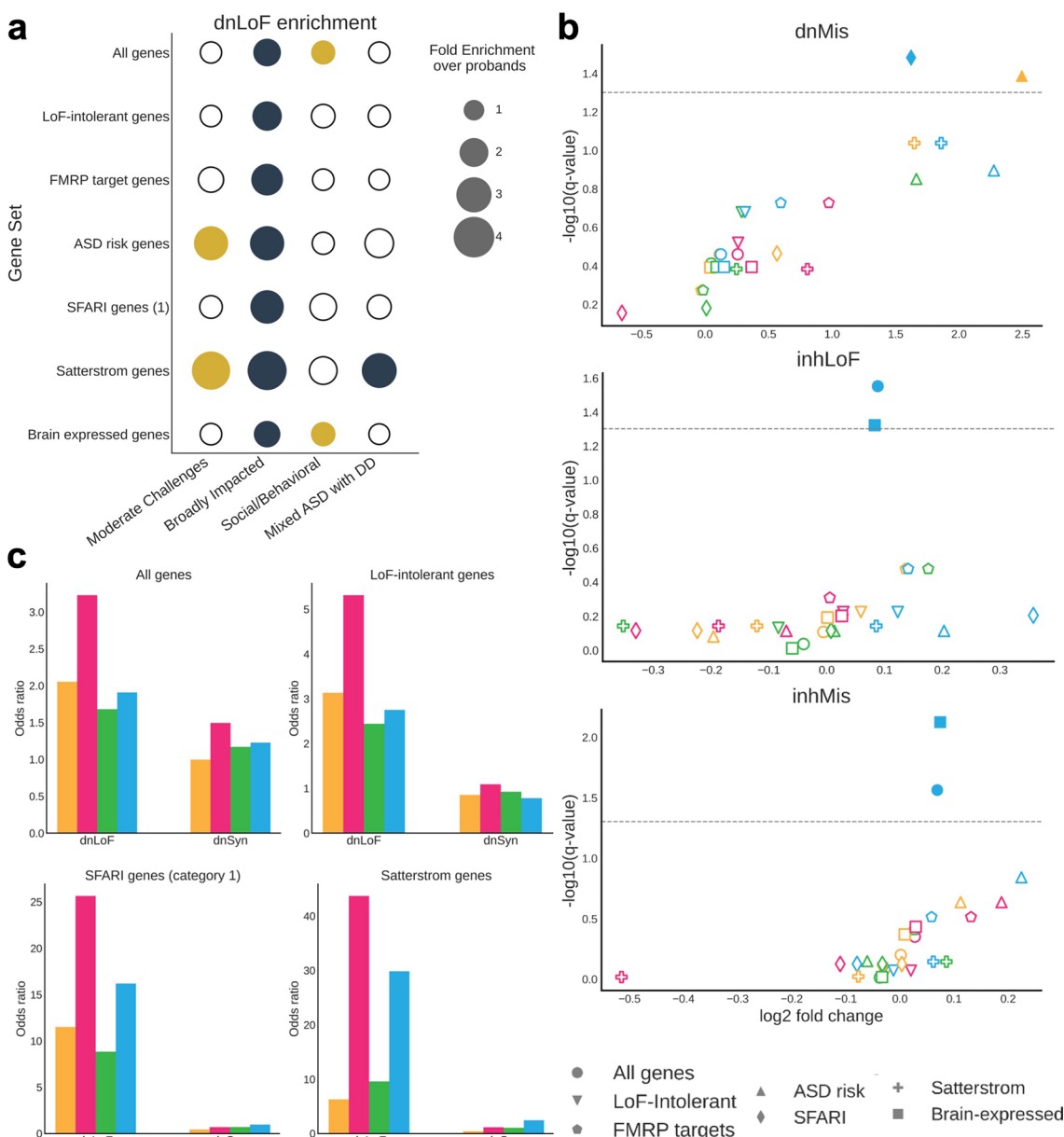

**Extended Data Fig. 8 | Gene set variant burden analyses. a**, dnLoF burden was computed for each class against out-of-class probands in each of seven gene sets (y-axis) using one-sided independent t-tests (adjusted for multiple comparisons using Benjamini-Hochberg correction). Fold enrichment (bubble size), FDR significance (closed vs. open circles at indicated threshold of 0.05), and direction of enrichment (color) are shown. Sample sizes in all analyses shown: Moderate challenges n = 809, Broadly affected n = 145, Social/behavioral n = 640, Mixed ASD with DD n = 419, unaffected siblings n = 1,013. All computed statistics, class comparisons, and comparisons to siblings can be found in Supplementary Table 8. **b**, Scatter plots displaying fold enrichment (x-axis) versus FDR significance (y-axis) of *de novo* missense (dnMis) burden, rare inherited LoF (inhLoF) burden,

and rare inherited missense (inhMis) burden across classes and seven autism-relevant gene sets (symbol key). We computed the aggregated burdens for each individual across every gene set. P-values and log₂ fold change were computed relatively to non-autistic siblings using one-sided independent t-tests. P-values were adjusted for multiple comparisons using Benjamini-Hochberg correction to compute the log-transformed Q-values (y-axis). Empty shapes represent tests that did not pass multiple hypothesis testing with a threshold of 0.05. Each gene set is represented by a different symbol as indicated in the key. **c**, Odds ratios for various gene sets included in this manuscript. Odds ratios (y-axis) were computed for dnLoF and dnSyn variants in each autism class against non-autistic siblings across autism-relevant gene sets (four shown here, two shown in Fig. 4b).

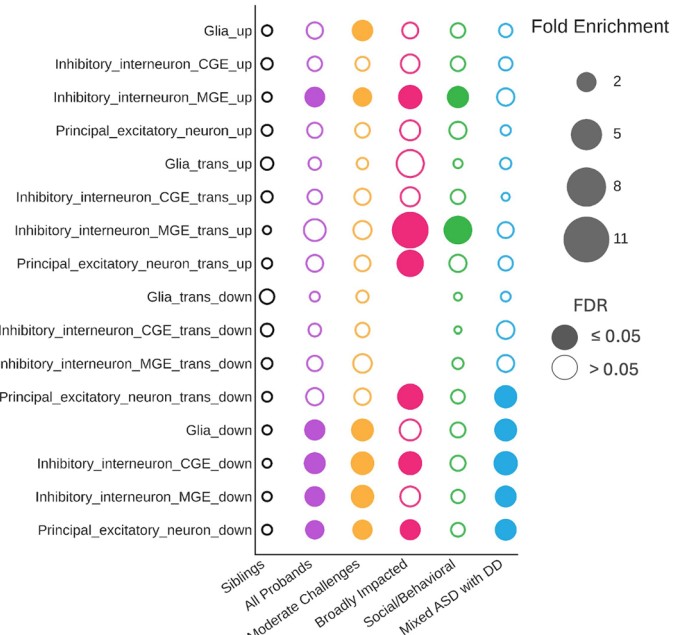

**Extended Data Fig. 9 | dnLoF enrichments across PFC celltypes, developmental gene trends, and phenotype classes.** dnLoF variant enrichment was computed across cell types and developmental gene expression trends (*y*-axis) for each phenotype class (*x*-axis). Open circles represent lack of significance (FDR > 0.05), whereas closed circles represent FDR ≤ 0.05 (one-sided independent *t*-tests adjusted for multiple comparisons with Benjamini-Hochberg correction). Lack of circles represents lack of data. Colors correspond to phenotype class colors and bubble size corresponds to fold enrichment. Sample sizes are: all probands (*n* = 2,013), Moderate challenges *n* = 809, Broadly affected *n* = 145, Social/behavioral *n* = 640, Mixed ASD with DD *n* = 419, unaffected siblings *n* = 1,013.

# Reporting Summary

## Statistics

For all statistical analyses, confirm that the following items are present in the figure legend, table legend, main text, or Methods section.

| n/a | Confirmed | |
|---|---|---|
| ☐ | ☒ | The exact sample size (*n*) for each experimental group/condition, given as a discrete number and unit of measurement |
| ☐ | ☒ | A statement on whether measurements were taken from distinct samples or whether the same sample was measured repeatedly |
| ☐ | ☒ | The statistical test(s) used AND whether they are one- or two-sided *Only common tests should be described solely by name; describe more complex techniques in the Methods section.* |
| ☐ | ☒ | A description of all covariates tested |
| ☐ | ☒ | A description of any assumptions or corrections, such as tests of normality and adjustment for multiple comparisons |
| ☐ | ☒ | A full description of the statistical parameters including central tendency (e.g. means) or other basic estimates (e.g. regression coefficient) AND variation (e.g. standard deviation) or associated estimates of uncertainty (e.g. confidence intervals) |
| ☐ | ☒ | For null hypothesis testing, the test statistic (e.g. *F*, *t*, *r*) with confidence intervals, effect sizes, degrees of freedom and *P* value noted *Give P values as exact values whenever suitable.* |
| ☐ | ☒ | For Bayesian analysis, information on the choice of priors and Markov chain Monte Carlo settings |
| ☐ | ☒ | For hierarchical and complex designs, identification of the appropriate level for tests and full reporting of outcomes |
| ☐ | ☒ | Estimates of effect sizes (e.g. Cohen's *d*, Pearson's *r*), indicating how they were calculated |

*Our web collection on statistics for biologists contains articles on many of the points above.*

## Software and code

Policy information about availability of computer code

| Data collection | No software was used to collect data. |
|---|---|
| Data analysis | HAT (no version number but downloaded from GitHub on 10/11/22) was used for variant calling.<br>plink v1.9 was used to compute polygenic scores.<br>ShinyGO 0.80 was used to perform GO term enrichment analyses.<br>StepMix 1.2.5 was retrieved from PyPI and used to construct and train the mixture models.<br>Ensembl VEP (Release 111.0) was used to call variant effects for rare coding variants.<br>LOFTEE (v.1.0.4) was used to call loss-of-function variants.<br>AlphaMissense (VEP plugin) was used to call missense variants.<br>DeepVariant (v1.1.0) was used for variant calling.<br>GATK HaplotypeCaller (v4.1.2.0) was used for variant calling. |

For manuscripts utilizing custom algorithms or software that are central to the research but not yet described in published literature, software must be made available to editors and reviewers. We strongly encourage code deposition in a community repository (e.g. GitHub). See the Nature Portfolio guidelines for submitting code & software for further information.

## Data

Policy information about availability of data

All manuscripts must include a data availability statement. This statement should provide the following information, where applicable:
- Accession codes, unique identifiers, or web links for publicly available datasets
- A description of any restrictions on data availability
- For clinical datasets or third party data, please ensure that the statement adheres to our policy

In order to abide by the informed consents that individuals with autism and their family members signed when agreeing to participate in a SFARI cohort (SSC and SPARK), researchers must be approved by SFARI Base (https://base.sfari.org).

## Research involving human participants, their data, or biological material

Policy information about studies with human participants or human data. See also policy information about sex, gender (identity/presentation), and sexual orientation and race, ethnicity and racism.

| Reporting on sex and gender | SPARK Consortium. Electronic address: pfeliciano@simonsfoundation.org & SPARK Consortium. SPARK: A US Cohort of 50,000 Families to Accelerate Autism Research. Neuron 97, 488–493 (2018). |
|---|---|
| Reporting on race, ethnicity, or other socially relevant groupings | SPARK Consortium. Electronic address: pfeliciano@simonsfoundation.org & SPARK Consortium. SPARK: A US Cohort of 50,000 Families to Accelerate Autism Research. Neuron 97, 488–493 (2018). |
| Population characteristics | The study cohort consisted of 5,392 children with a professional autism diagnosis aged 4-18 years (mean = 8.56 years, SD = 3.15). The control population consisted of 1,972 siblings without a professional diagnosis of autism aged 4-18 years (mean = 7.95 years, SD = 4.41). In these populations, 4,636 probands and 1,972 siblings had whole exome sequencing (WES) data available.<br><br>Additional details can be found in: SPARK Consortium. Electronic address: pfeliciano@simonsfoundation.org & SPARK Consortium. SPARK: A US Cohort of 50,000 Families to Accelerate Autism Research. Neuron 97, 488–493 (2018). |
| Recruitment | SPARK Consortium. Electronic address: pfeliciano@simonsfoundation.org & SPARK Consortium. SPARK: A US Cohort of 50,000 Families to Accelerate Autism Research. Neuron 97, 488–493 (2018). |
| Ethics oversight | We received approval to access and analyze de-identified genetic and phenotypic data from the two cohorts from SFARI Base and the Princeton University IRB Committee in the Office of Research Integrity. |

Note that full information on the approval of the study protocol must also be provided in the manuscript.

# Field-specific reporting

Please select the one below that is the best fit for your research. If you are not sure, read the appropriate sections before making your selection.

☒ Life sciences  ☐ Behavioural & social sciences  ☐ Ecological, evolutionary & environmental sciences

For a reference copy of the document with all sections, see nature.com/documents/nr-reporting-summary-flat.pdf

# Life sciences study design

All studies must disclose on these points even when the disclosure is negative.

| Sample size | n = 5,392 (SPARK proband cohort), n = 1,972 (SPARK sibling cohort), and n = 861 (SSC proband cohort). Measures in SPARK with sufficiently high overlap among participants were selected to maximize the cohort size while maintaining breadth of phenotype data. All probands with complete or mostly complete data for four phenotype battery assays (Background History, Social Communication Questionnaire, Repetitive Behavior Scale, and Child Behavior Checklist) were included in the analysis. More details are available in the Methods section. Extensive statistical testing for both phenotypic and genetic analyses, including adjustment for multiple hypotheses, validated the sufficiency of these sample sizes. |
|---|---|
| Data exclusions | Data exclusion was performed based on availability and completion of data. Every participant who had the required data available (mostly complete across four phenotype battery assays) was included. Additionally, for genetic analyses of de novo variation, we excluded individuals who had an outlier count of de novo variants (defined as 3 standard deviations above the mean de novo count across the sample). |
| Replication | Replication was performed on a phenotype dataset from another cohort (the Simons Simplex Collection) which included n = 861 participants with complete phenotype information for four phenotype assays. Multiple replication methods were tested and all replication attempts were successful. More details on replication are available in the Methods section. |
| Randomization | Covariates were controlled for in the main mixture model used to classify the sample. Covariates include sex and age at evaluation as |

| Randomization | provided by the phenotype dataset. |
|---|---|
| Blinding | This study is a computational analysis of data-drive subclasses of autism. We received approval to analyze SPARK and SSC data from SFARI Base, and we had no contact with study participants. No treatment was administered, and therefore blinding is not applicable to this line of data-driven analysis. |

# Reporting for specific materials, systems and methods

We require information from authors about some types of materials, experimental systems and methods used in many studies. Here, indicate whether each material, system or method listed is relevant to your study. If you are not sure if a list item applies to your research, read the appropriate section before selecting a response.

## Materials & experimental systems

| n/a | Involved in the study |
|---|---|
| ☒ ☐ | Antibodies |
| ☒ ☐ | Eukaryotic cell lines |
| ☒ ☐ | Palaeontology and archaeology |
| ☒ ☐ | Animals and other organisms |
| ☒ ☐ | Clinical data |
| ☒ ☐ | Dual use research of concern |
| ☒ ☐ | Plants |

## Methods

| n/a | Involved in the study |
|---|---|
| ☒ ☐ | ChIP-seq |
| ☒ ☐ | Flow cytometry |
| ☒ ☐ | MRI-based neuroimaging |

## Plants

| Seed stocks | *Report on the source of all seed stocks or other plant material used. If applicable, state the seed stock centre and catalogue number. If plant specimens were collected from the field, describe the collection location, date and sampling procedures.* |
|---|---|
| Novel plant genotypes | *Describe the methods by which all novel plant genotypes were produced. This includes those generated by transgenic approaches, gene editing, chemical/radiation-based mutagenesis and hybridization. For transgenic lines, describe the transformation method, the number of independent lines analyzed and the generation upon which experiments were performed. For gene-edited lines, describe the editor used, the endogenous sequence targeted for editing, the targeting guide RNA sequence (if applicable) and how the editor was applied.* |
| Authentication | *Describe any authentication procedures for each seed stock used or novel genotype generated. Describe any experiments used to assess the effect of a mutation and, where applicable, how potential secondary effects (e.g. second site T-DNA insertions, mosiacism, off-target gene editing) were examined.* |

