## [Peer Review File · Nature Genetics]

Decomposition of phenotypic heterogeneity in autism reveals underlying genetic programs

Corresponding Author: Dr Olga Troyanskaya

Version 0:

Decision Letter:

20th September 2024

Dear Olga,

Your Article "Decomposition of phenotypic heterogeneity in autism reveals distinct and coherent genetic programs" has been seen by three referees. You will see from their comments below that, while they find your work of interest, they have raised substantial concerns that must be addressed. In light of these comments, we cannot accept the manuscript for publication at this time, but we would be interested in considering a suitably revised version that addresses the referees' concerns.

We hope you will find the referees' comments useful as you decide how to proceed. If you wish to submit a substantially revised manuscript, please bear in mind that we will be reluctant to approach the referees again in the absence of major revisions.

To guide the scope of the revisions, the editors discuss the referee reports in detail within the team, including with the chief editor, with a view to identifying key priorities that should be addressed in revision, and sometimes overruling referee requests that are deemed beyond the scope of the current study. In this case, we ask that you rigorously address all technical queries related to the phenotypic clustering by exploring alternate models and performing further analyses to assess their stability and generalizability, extend the genetic analyses to include direct comparisons between case subgroups, and revise the interpretations where needed taking all referee comments into account. We hope you will find this prioritized set of referee points to be useful when revising your study. Please do not hesitate to get in touch if you would like to discuss these issues further.

If you choose to revise your manuscript taking into account all reviewer and editor comments, please highlight all changes in the manuscript text file. At this stage, we will need you to upload a copy of the manuscript in MS Word .docx or similar editable format.

*2) If you have not done so already, please begin to revise your manuscript so that it conforms to our Article format instructions, available here. Refer also to any guidelines provided in this letter.

*3) Include a revised version of any required Reporting Summary: <https://www.nature.com/documents/hr-reporting-summary.pdf>

Please be aware of our [guidelines](https://www.nature.com/nature-research/editorial-policies/image-integrity) on digital image standards.

Link Redacted

If you wish to submit a suitably revised manuscript, we hope to receive it within 3-6 months. If you cannot send it within this time, please let us know. We will be happy to consider your revision so long as nothing similar has been accepted for publication at Nature Genetics or published elsewhere. Should your manuscript be substantially delayed without notifying us in advance and your article is eventually published, the received date would be that of the revised, not the original, version.

Nature Genetics is committed to improving transparency in authorship. As part of our efforts in this direction, we are now requesting that all authors identified as 'corresponding author' on published papers create and link their Open Researcher and Contributor Identifier (ORCID) with their account on the Manuscript Tracking System (MTS), prior to acceptance. ORCID helps the scientific community achieve unambiguous attribution of all scholarly contributions. You can create and link your ORCID from the home page of the MTS by clicking on 'Modify my Springer Nature account'. For more information, please visit www.springernature.com/orcid.

Thank you for the opportunity to review your work.

Sincerely,
Kyle

Kyle Vogan, PhD
Senior Editor
Nature Genetics
<https://orcid.org/0000-0001-9565-9665>

Referee expertise:

Referee #1: Genetics, autism, computational modeling

Referee #2: Genetics, autism, statistical methods

Referee #3: Genetics, autism, statistical methods

Reviewers' Comments:

Reviewer #1:
Remarks to the Author:

Heterogeneity in autism is a serious and important challenge to be addressed. Against this background, Littman and colleagues have taken an interesting approach by combining finite mixture modelling alongside genetics to understand heterogeneity. Although finite mixture modelling has been conducted before in autism, the novelty here is two-fold - a large number of phenotypes included in the model, and comparing the phenotypic (latent) classes against a genetic background. Although this approach is of interest, there are a few aspects of the manuscript that I have some concerns about, which I outline below.

1. To me, there is no clear empirical support as to why the authors chose four classes. I agree that there is subjectivity involved with these decisions, but there are aspects in here that are unsatisfactory. Specifically, when looking at the fit statistics (AIC, BIC, CAIC, SABIC), they keep dropping as the number of latent classes increases. However, the most significant drop (a drop > 10 , which is typically used as a good benchmark of strong support for one model over the other) is when going from 1 latent class to 2. The other aspects - entropy, elbow plots - all provide contradictory information, making it difficult to clearly say that there genuinely are four latent classes. One fundamental reason for this could be the feature selection steps prior to running the mixture models. Given the high correlations among items within, say, the SCQ or the CBCL, I would recommend removing highly correlated variables and checking for model convergence.

2. Throughout the manuscript, the authors have investigated if the four classes of autistic individuals have elevated rates of genetic variants compared to sibling controls. However, to formally make statements like this: "In addition to distinct patterns

of common variants, we also observed significant differences in rare genetic variation between the phenotypic classes.”, the authors need to ideally test if genetic variants differ among the four classes, rather than just when compared to sibling controls. The authors are requested to run these to empirically support their statements of significant group differences.

3. One of the difficulties of mixture models in autism is that, beyond the initial cohort in which they were constructed, they are not necessarily widely generalizable to other cohorts. To this end, it was very nice to read efforts to replicate the findings in the SSC. However, I would like to see more statistics about this replication. For instance, what were the fit statistics of the model trained on SPARK using the 108 features? Did it still identify a four-class solution as the best one? The authors have fitted the trained model in SPARK onto SSC. However, to truly evaluate the generalisability, it would be better to train a new model in SSC using exactly the same features used SPARK and SSC, and compare the identified models.

4. I found figure 4a difficult to follow - please, can the authors try a different way of presenting the same information?

5. Supplementary Table 7 and section on developmental gene expression - it does not look like any of the cell type/developmental gene expression tests survive multiple testing correction. Given this, I find it difficult to evaluate to what extent the results in the developmental gene expression are true positives.

Reviewer #2:

Remarks to the Author:

In “Decomposition of phenotypic heterogeneity in autism reveals distinct and coherent genetic programs,” Litman and colleagues identify groups of autistic individuals through decomposition of SPARK’s phenotypic structure, and note similar structure in the SSC dataset. They then link the classes to average differences in genetic architecture. There are many benefits to such an analysis, including identification of phenotypic clusters and potential to understand average differences in their etiological architecture. The analyses must be interpreted carefully and consider what has already been established within the field. I hope these comments are helpful in continuing the research.

1. I think the paper would benefit from removal of the word “distinct,” which oversells observed differences – phenotypic and genetic – between the classes. The term distinct, to many, implies a bucketing sort of phenomenon (a and b don’t meaningfully overlap), which doesn’t seem to be the sense in which it is used here. Phenotypically, it isn’t clear for which measures variance between classes exceeds variance within classes. More generally, phenotypic variation within the classes isn’t made clear, and that seems important. Genetically, the groups are highly overlapping in terms of their etiologic architectures. Compared to a non-autistic population, are they more similar (overlapping) than distinct? A similar concern extends to the term “coherent” as used in the title and perhaps other places in the paper. I do not think the data presented support either “distinct” or “coherent”.

2. The phenotypic clusters are interesting, though it should be noted they likely reflect many potential points of origin, particularly rater effects. Autistic individuals with limited speech and/or global developmental delays/intellectual disability (ID) have very different phenotypic profiles as rated, as one would expect. They are less likely to be accurately assessed for anxiety and mood symptoms, and quite naturally have different average profiles in terms of social behavior. The extent to which the four classes that emerged track with global developmental delay is not adequately considered or discussed in the manuscript. All of the measures employed in the class identification activity have established associations to ID/IQ, and the classes here largely recapitulate expected relationships between ID/IQ and behavioral phenotypes in autism.

3. Given the clear role of ID/IQ in formation of the classes, the genetic differences observed are largely expected. The class in which the largest fraction of individuals meet criteria for intellectual disability should have the highest rate of strong acting de novo events, as observed here. Further, observed mutations in that group are expected to reside with genes expressed earliest in development. A suite of papers over the last several years (e.g. Fu et al. 2023) have shown average difference in expression timing between genes associated with ID, autism and schizophrenia. Within autism-associated genes, those also associated to ID are expressed earlier than those not. Genetic variants most disruptive to human brain development are found in genes expressed earliest, and those associated to ID are more likely to fall within the FMRP pathway. We therefore expect the class of individuals with the greatest ID concentration to show those genetic patterns [note: given the ID/IQ data available in SPARK, greatest burden of global developmental delay is best indicated by milestone delay data.] Additionally, we expect autistic individuals with more severe ID to have a mixture of challenges – as rated – because the presence of ID and autism makes many behavioral phenotypes difficult to assess. On the other hand, we expect individuals in the class with the lowest ID concentration (highest average IQ) to have the lowest rate of autism/ID-associated disruptive variation, and we expect the genes in which the variants they do have are found to be expressed later. We expect the highest IQ group to have the highest rate of anxiety, along with challenges more common in autistic individuals with higher IQ. We expect the groups in between to have a mixed genetic architecture, including an elevated rate of rare inherited, putatively damaging variation. The same expectation logic extends to the observed polygenic score differences, with a note that PGS provide clear opportunity to compare variation within versus between the classes, and that analysis would be of benefit.

4. It is statistically challenging to compare pathway enrichment, or rare variant rate enrichment, particularly at this sample size. Perhaps I missed it, but it seemed the authors identified enrichment within each group, as compared to null (control expectation). They did not statistically compare enrichment between the groups? This would require multiple testing correction, and apologies if I missed this somewhere.

Reviewer #3:

Remarks to the Author:

Paper Summary:

Litman, Sauerwald and colleagues took aim at phenotypic heterogeneity within autism. They used an unsupervised, person-centered clustering approach in the SPARK cohort (N of ~5,000) to identify four groups of patients with intuitive interpretability – moderate challenges, broad impact, social/behavioral, and mixed ASD/DD. They suggest these clusters to be ‘significant’ on the basis of goodness of fit metrics, interpretability, and external validity via projection into a smaller SSC cohort (N of ~900). From there, they investigate the genetic architecture of these clusters using a set of polygenic scores and burden of rare de novo and inherited variation. The rare variant burdens were further interpreted for pathway and gene expression signatures. The paper is interesting, creative, pursuing an important question, and well written – clearly, concisely explained and motivated. The methods are clearly explained and visualizations both clean and interpretable. However, despite my enthusiasm for the approach, I have some concerns as to the strength of the evidence supporting the claim of ‘distinct genetic programs.’ These center around wanting a bit more demonstration to the stability of the clusters and how the tests used to identify genetic differences among clusters were performed.

Major Comments:

How stable and robustly separable are the clusters?

I understand focusing downstream analysis on a best solution, but I would like more intuition for how stable and separable these particular clusters are. There is emerging evidence (as cited by the authors) that aspects of clinical presentation relate to aspects of genetics. As such, would any clustering solution produce genetic differences that could be justified post hoc? I think (and hope) not. I also like the ‘person-centered’ approach - interesting and feels like a ‘right’ philosophical approach. However, I’d like a bit more demonstration that these particular clusters are ‘special’ / represent some ‘natural kind’ / don’t just trim axes of quantitative covariation.

The goodness of fit metrics use an ‘elbow rule’ heuristic to select four clusters, but the separation of those indices seems more ‘quantitative’ than ‘qualitative.’ It appears (correct me if I am wrong) that all of the metrics for goodness of fit were based on an analysis of the full sample. Does a resampling or subsampling based approach suggest the four clusters are stable to small perturbations of the data? Are subjects frequently flipping classes depending on the context? What is the feature concordance among re- or sub-samples, as tested with the replication? How stable is the feature enrichment to re- or sub-sampling?

Interpretability was used as one metric to choose four clusters, but the metrics for comparator solutions are not in the paper. Four clusters had the ‘best phenotypic separation and most clinically coherent classes’ but it would be useful to see that data. Could the authors provide, e.g., metrics as in Figure 1 for a few candidate models – 3-6 clusters?

How does the GFMM handle potential confounding covariates? It is mentioned “1-step StepMix with age and sex as covariates” – how are these treated by the clustering? What about genetic ancestry? It is a little unclear if a multi- or single-ancestry sample was used for the clustering. If multi-ancestry, this can be problematic as associated social factors could impact clustering. Can PCs be adjusted or ancestry tested in other ways for impact on clustering?

For the SSC replication, is shuffling the test sample labels the appropriate null? Would it not make more sense to shuffle the labels of the SPARK, project those clusters into the SSC, and test concordance? i.e., is the concordance between SPARK and SSC of the real cluster projection better than a permuted cluster projection? This would better support stability of those particular clusters, no?

How much additional support do the BMS and parent/self-report give? Is their enrichment not tautological? Many of the outcomes used to confirm the clusters are ‘naturally’ related to the instruments used to construct the clusters (CBCL -> BMS; BhX -> parent/self-report milestones), so are we validating the clusters or the construct validity of instruments (i.e., ‘validating’ height in inches with height in centimeters)? I am not sure how to address this, but it is worth discussing/tempering this additional evidence that the clusters are ‘phenotypically distinct.’ This issue aside, the enrichments are reported/tested against the non-autistic siblings, not in a competitive way against the classes, so we cannot know if the differences in enrichment among groups (the key concept) is significant, as it is not reported.

Evidence for genetic differences

As I understand, the significance of (all of) the genetic enrichments stems from differences in dichotomized significance of case-control tests, i.e., sub-group A is significantly different from controls, but sub-group B is not. These kinds of inferences are problematic and can represent false dichotomies/winner’s curse biases. A formal test of the difference in the case-control enrichments or difference between the case sub-groups, directly, is (at minimum) needed to justify the claims of distinct genetic programs. A pessimistic read suggests nearly all of the subgroup genetic enrichments are not significant between case-groups, given the nearly overlapping standard errors (where provided). I believe this affects results from Figures 3A-C, 4A-C, and 5B.

Rather than contrasting significance levels (which can be driven by differences in the size of the groups), it would be more useful to contrast effect sizes, and to use 95% CI instead of SEM (easier to intuit subgroup differences as meaningful). When an enrichment is proposed for one sub-group, it would be useful to show its level and 95% for all other subgroups (not done in panel 4C).

Some of these “enrichment tests” (4C, 5B) use complicated heuristic statistics and may not have intuitive effect sizes with easy estimates of variance, and resampling procedures could be complicated. I am less worried about these more descriptive analysis (although see my comment about 4C above), but for 3A-C and 4A-B, in particular, I think the sub-group differences should be more rigorously tested.

Minor Comments:

For genetic analysis, it mentions self-report European, but this can/should be confirmed with genetic ancestry PCs.

ASD has increased in prevalence and breadth over recent decades. Some previous subgroups are gone (Asperger's) and some individuals previous categorized as intellectual disability or developmental delay may now fall in ASD. Could the authors discuss/speculate if their clusters are ‘rediscovering’ previous clinical groups? Is the move to a more ‘inclusive ASD’ perhaps not supported by etiological data?

Is this use of non-autistic siblings as a control group the most efficient? From my read, non-autistic siblings, as a pool, are used to estimate some aspects of enrichment (Figure 3A), and not via a paired analysis. I wonder if this doesn't mix apples with oranges. Take sibships A1-A2 and B1-B2 (A1/B1 case, and A2/B2 control). The differences in PGS between A1-A2 and B1-B2 represent within family genetic variance (segregation variance, roughly half the total genetic variance) while the difference between A1-B2 and B1-A2 represent both within and between family genetic variance (the full genetic variance). The genetic distance (e.g., mean PGS difference) between cases and controls is then a mixture of within and between family differences, with within family differences expected to be smaller). In other words, we expect the PGS of the ASD sibs to be closer to their non-ASD sib than another cases non-ASD sib. Could this affect the PGS (or rare variant?) enrichments? Differences between cases and controls for larger subgroups will be more weighted towards within family differences, while smaller groups to between family differences, as larger subgroups will have more sibs in the controls, smaller subgroups, fewer sibs. I wonder if this mixing of within and between family contrasts doesn't add noise to the enrichment tests. Case-subgroup contrasts would not suffer this – A1-B1 differences capture differences in the full genetic variance (within and between family).

Is parental history of disorders available? This could be an independent replication of genetic differences if the clusters had parents enriched for the same disorders as they have enriched PGS. It turns out these can be relatively independent measure of genetic liability (e.g., PMID: 36347255).

Version 1:

Decision Letter:

Our ref: NG-A66184R

27th March 2025

Dear Olga,

Your revised manuscript "Decomposition of phenotypic heterogeneity in autism reveals distinct and coherent genetic programs" (NG-A66184R) has been seen by Reviewer #1. (Reviewers #2 and #3 were also asked to comment on the revision but have not returned comments to date.) In light of Reviewer #1's comments, we will be happy in principle to publish your study in Nature Genetics as an Article pending final revisions to satisfy Reviewer #1's remaining requests and to comply with our editorial and formatting guidelines.

We are now performing detailed checks on your paper, and we will send you a checklist detailing our editorial and formatting requirements soon. Please do not upload the final materials or make any revisions until you receive this additional information from us.

Thank you again for your interest in Nature Genetics. Please do not hesitate to contact me if you have any questions.

Sincerely,
Kyle

Kyle Vogan, PhD
Senior Editor
Nature Genetics
<https://orcid.org/0000-0001-9565-9665>

Reviewer #1 (Remarks to the Author):

The authors have done a great job in addressing many of my comments. This is an important concept to address and interrogate. Upon re-reading the reviews and the paper, however, I do agree with reviewer 2's first point (which is the spirit behind my first point about the number of classes). I do think that distinctive is not the right word over here for a few reasons. First, if I were to look at the elbow plots and the fit statistics (I still don't see an elbow, I see a gentle curve), I would pick out a two-class solution as the best one. This may likely reflect a separation based on ID/DDD. It would still make sense clinically. I don't think the authors are incorrect, and I'm merely highlighting the subjectivity of this process. Second, although there is greater variance within than between, there isn't a clear genetic or phenotypic separation between the groups to use the term distinctive. The interested but non-specialist reader would walk away with the message that there are four clear, separate groups among autistic individuals. I don't think this is what the manuscript is saying (and I don't think this is what the authors intend to suggest). Rather, I think the manuscript points to certain dimensions in a multivariate phenotypic space that can be used to cluster autistic individuals into groups. These groups differ in their genetic profile. A litmus test would be to ask if you see an autistic child with a combination of phenotypes can you easily classify them into one of four groups? I suspect not.

So, I would strongly emphasise against the use of the word distinctive (how about "differing" or "dissimilar" instead?). I'm also not sure about the word coherent - what do you mean by coherent here? Do you mean that it comes together nicely? Apologies, I don't seem to get it (and I'm not sure if it's needed here).

So, in sum, I think the authors have done a great job addressing my comments. I think this is an important contribution in the ongoing debate on the optimal way to characterise autism. That said, in the spirit of the debate, I would recommend using a different word from distinct and not using the term coherent as these terms belie the subjective and fuzzy nature of these groupings.

Reviewer #1:

Remarks to the Author:

Heterogeneity in autism is a serious and important challenge to be addressed. Against this background, Littman and colleagues have taken an interesting approach by combining finite mixture modelling alongside genetics to understand heterogeneity. Although finite mixture modelling has been conducted before in autism, the novelty here is two-fold - a large number of phenotypes included in the model, and comparing the phenotypic (latent) classes against a genetic background. Although this approach is of interest, there are a few aspects of the manuscript that I have some concerns about, which I outline below.

We appreciate the reviewer's careful consideration of our manuscript, and their valuable suggestions for improvement.

1. To me, there is no clear empirical support as to why the authors chose four classes. I agree that there is subjectivity involved with these decisions, but there are aspects in here that are unsatisfactory. Specifically, when looking at the fit statistics (AIC, BIC, CAIC, SABIC), they keep dropping as the number of latent classes increases. However, the most significant drop (a drop > 10 , which is typically used as a good benchmark of strong support for one model over the other) is when going from 1 latent class to 2. The other aspects - entropy, elbow plots - all provide contradictory information, making it difficult to clearly say that there genuinely are four latent classes. One fundamental reason for this could be the feature selection steps prior to running the mixture models. Given the high correlations among items within, say, the SCQ or the CBCL, I would recommend removing highly correlated variables and checking for model convergence.

We understand and appreciate the concern about selecting the best number of classes for the mixture model; this is not a straightforward process with mixed guidance in the literature, though the general consensus emphasizes a thorough examination of a host of fit metrics alongside practical interpretation [Little, 2013, Nylund-Gibson and Choi, 2018, Sinha et al, 2020].

Sources suggest using the number of classes at which returns in statistical metrics such as the LL, BIC, and AIC are diminishing (often called an "elbow"), rather than the number with the largest drop [Little, 2013]. "In practice, it is not uncommon that the [LL, BIC, AIC, etc] continues to decrease for each additional class added (e.g., there is not a global minimum) and in these instances these plots can be particularly useful to inspect for an 'elbow' of point of 'diminishing returns' in model fit" [Nylund-Gibson and Choi, 2018]. Based on this guidance, we selected a 4-class rather than a 2-class model to match the elbow criteria.

Literature on mixture modeling also emphasizes the importance of not just the statistical metrics but also the interpretability of the selected model, recommending "substantive scrutiny and practical reflection" [Little, 2013, Nylund-Gibson and Choi, 2018, Sinha et al, 2020]. To this end, prior to any downstream validation and analyses, we presented

models with 2-6 classes to clinical collaborators with extensive experience working with autistic children. The four class model most accurately captured realistic subclasses of autism, with characteristics within classes that clinicians had observed in their practice, and differences across classes that were meaningful and represented differences across children they had worked with. We have added the following text to the manuscript to better reflect this process of model selection:

Main text: We selected a GFMM with four latent classes representing four distinct patterns of phenotype profiles **by considering six standard model fit statistical measures and overall interpretability of the resulting model**. After training models with 2-10 latent classes, we found that four classes presented the best balance of model fit as measured by the Bayesian Information Criterion (BIC), Validation Log Likelihood, and other statistical measures of fit (Supplementary Fig. 1; Supplementary Table 1). Additionally, a four class solution offered the best interpretability in terms of phenotypic separation, as evaluated by clinical collaborators with extensive experience working with autistic individuals (Methods). **Characteristics of 3-, 5- and 6-class models can additionally be seen in Supplementary Figure 2.**

Methods: The main tunable hyperparameter of the GFMM is the number of latent classes. **It is recommended that this number is chosen by identifying an “elbow” in statistical measures of model fit, in combination with careful consideration of practical interpretability¹⁻³.**

Additionally, we want to thank the reviewer for the suggestion about checking the model's robustness to the correlated variables. Based on this suggestion, we have implemented a systematic process to remove highly correlated variables ($\text{abs}(\text{Pearson } r) > 0.5$) within each variable type (binary, categorical, continuous), and found that training the model with uncorrelated variables results in similar statistical measures and interpretations as the full model (Figure R1.1). The correlation between the full 4-class model and the 4-class model generated from uncorrelated features is 0.9329.

Figure R1.1: Statistical metrics for 1-6 class models trained only with uncorrelated features (left) and for 1-10 class models trained with the full feature set (right).

2. Throughout the manuscript, the authors have investigated if the four classes of autistic individuals have elevated rates of genetic variants compared to sibling controls. However, to formally make statements like this: “In addition to distinct patterns of common variants, we also observed significant differences in rare genetic variation between the phenotypic classes.”, the authors need to ideally rest if genetic variants differ among the four classes, rather than just when compared to sibling controls. The authors are requested to run these to empirically support their statements of significant group differences.

Thank you for the suggestion; we have now run statistical testing for all differences across groups, and reported FDR values in the text, as well as marked many of these relationships in the figures (Figures 1C, 1D, 2B, 3A-C). In order to maintain visual clarity, many of these have been added to the supplement rather than shown in figures. We have also added Supplementary Figure 11 to portray gene set comparisons within the pool of probands. The statistics were additionally computed for all comparisons in Figures 2A, 4A, and 5B. We report the full results of these comparisons in updated and new Supplementary Tables 2, 3, 4, 6, 7, 8, and 12. Notably, the analysis in Figure 4C does not include any comparisons to siblings as a control group; this is a gene-set based

analysis rather than person-based enrichment, and therefore does not lend itself to statistical testing between case-groups. Additionally, all relevant figures and tables were also updated with the latest whole exome data, released by SPARK in August 2024 (n=414 new exomes for our cohort).

3. One of the difficulties of mixture models in autism is that, beyond the initial cohort in which they were constructed, they are not necessarily widely generalizable to other cohorts. To this end, it was very nice to read efforts to replicate the findings in the SSC. However, I would like to see more statistics about this replication. For instance, what were the fit statistics of the model trained on SPARK using the 108 features? Did it still identify a four-class solution as the best one? The authors have fitted the trained model in SPARK onto SSC. However, to truly evaluate the generalisability, it would be better to train a new model in SSC using exactly the same features used SPARK and SSC, and compare the identified models.

Replication is certainly a challenging and important consideration for this work. As the reviewer suggested, we have now plotted the fit statistics of the 108-feature model on SPARK (Figure R1.2), which show very similar patterns to those of the original 239-feature model (Supplementary Figure 1):

Figure R1.2: Statistical metrics for 1-6 class models trained with 108 features matched in the SSC.

Based on the reviewer's suggestion, we have also trained a new independent model in SSC using these same 108 features. Correlations across each phenotypic category with the full SPARK model are shown here (new Supplementary Figure 8), displaying high concordance despite the fact that less than half of the same phenotypic features were available for the SSC cohort.

Supplementary Figure 8: Correlation of feature enrichment patterns across 7 phenotype categories between the full SPARK model and an independently-trained SSC model.

4. I found figure 4a difficult to follow - please, can the authors try a different way of presenting the same information?

We apologize for the confusion. Based on the reviewer's suggestion, we have now changed the visualization to a bubble plot similar to Figure 5B, which can be seen below on the left (Figure R1.3 left, also main Figure 4A). Each row now represents a different gene set, each column represents a different phenotypic class, and circle size indicates fold enrichment relative to siblings. We have also added an additional figure to the supplement (Figure R1.3 right, also new Supplementary Figure 11), directly representing the differences between classes by showing the enrichments of these gene sets in a given class versus probands from all other classes, where the color indicates either enrichment or depletion. All class-class comparison statistics can also be found in Supplementary Table 8. We thank the reviewer for their helpful suggestion, which greatly clarified the presentation of this result.

Main Fig. 4A

Supplementary Fig. 11

Figure R1.3: Revised Main Figure 4A (left) displaying enrichment of dnLoF variation in each class compared to non-autistic siblings across 7 different relevant gene sets (y-axis). New Supplementary Figure 11 (right) displaying enrichment of dnLoF variation in each class compared to out-of-class probands across 7 different relevant gene sets. Fold enrichment (bubble size), FDR significance (open vs. closed circles at indicated thresholds), and color (class color on the left, enrichment or depletion on the right) are displayed.

5. Supplementary Table 7 and section on developmental gene expression - it does not look like any of the cell type/developmental gene expression tests survive multiple testing correction. Given this, I find it difficult to evaluate to what extent the results in the developmental gene expression are true positives.

We appreciate the reviewer's question. All values presented in the table and figure were corrected for multiple comparisons using a Benjamini-Hochberg correction, and an FDR cutoff of 0.05 was used to determine significance. Supplementary Table 7 has now been moved to Supplementary Table 12, and previously listed the FDR values under the header of "-log10 p-val", but in fact represented the -log10 FDR. We have uploaded a corrected version of the table with the proper header, and also included the uncorrected -log10 p-values for reference. We thank the reviewer for pointing this out, and hope that this alleviates concerns about false discoveries in the analysis.

Reviewer #2:

Remarks to the Author:

In “Decomposition of phenotypic heterogeneity in autism reveals distinct and coherent genetic programs,” Litman and colleagues identify groups of autistic individuals through decomposition of SPARK’s phenotypic structure, and note similar structure in the SSC dataset. They then link the classes to average differences in genetic architecture. There are many benefits to such an analysis, including identification of phenotypic clusters and potential to understand average differences in their etiological architecture. The analyses must be interpreted carefully and consider what has already been established within the field. I hope these comments are helpful in continuing the research.

We thank the reviewer for their insights and consideration of our manuscript.

1. I think the paper would benefit from removal of the word “distinct,” which oversells observed differences – phenotypic and genetic – between the classes. The term distinct, to many, implies a bucketing sort of phenomenon (a and b don’t meaningfully overlap), which doesn’t seem to be the sense in which it is used here. Phenotypically, it isn’t clear for which measures variance between classes exceeds variance within classes. More generally, phenotypic variation within the classes isn’t made clear, and that seems important. Genetically, the groups are highly overlapping in terms of their etiologic architectures. Compared to a non-autistic population, are they more similar (overlapping) than distinct? A similar concern extends to the term “coherent” as used in the title and perhaps other places in the paper. I do not think the data presented support either “distinct” or “coherent”.

We appreciate the reviewer’s comments, but respectfully disagree. The four classes, while of course sharing some characteristics as they represent individuals with the same overarching diagnosis, are both phenotypically and genetically significantly different. This is clearly demonstrated by the different phenotypic measures (Figures 1B, 1C, 1D, 2A, and 2B) and genetic differences (Figures 3A-C, 4A-C, 5B) we documented; we now include systematic statistical comparisons to further support the distinctions between these classes (Supplementary Tables 2, 3, 4, 6, 7, 8, 10, and 12).

We have also performed analyses to compare phenotypic variation between classes to phenotypic variation within classes (new Supplementary Figure 5). We found that within-class feature variation is significantly and substantially smaller than between-class feature variation ($p = 9.97e-176$, Supplementary Fig. 5, between-class distribution vs. within-class distribution), further supporting the coherence within and distinction between the classes.

Supplementary Figure 5: Comparison of feature variances between versus within classes.

We have also added the following text to the manuscript:

Results: Furthermore, classes display significant differences across measures (Supplementary Table 2), and significantly greater between-class variability than within-class variability (Supplementary Fig. 5), further supporting their phenotypic distinctiveness.

Furthermore, these classes demonstrate different genetic signatures with respect to impacted pathways (Figure 4C) and common (Figure 3A), rare *de novo* (Figures 3B, 4A, and 5B), rare inherited (Figure 3B), and evolutionarily constrained variants (Figure 3C). We also now directly tested significance of these differences and marked these in Figures 3A-C, clearly showing that the four classes indeed demonstrate significant genetic differences.

For your reference, we also include Table R2.1 to further summarize basic phenotypic and genetic findings for each class.

	Moderate Challenges	Broadly Impacted	Social/Behavioral	Mixed ASD with DD
Internal Phenotypes	 Low DD Low anxiety/mood, attention, disruptive, self-injury, RRB, SO/CO 	 High DD High anxiety/mood, attention, disruptive, self-injury, RRB, SO/CO 	 Low DD High anxiety/mood, attention, disruptive, self-injury, RRB, SO/CO 	 High DD Low anxiety/mood, attention, disruptive Mixed self-injury, RRB, SO/CO
External Phenotypes	 Low ID Language delay Low interventions 	 High ID High ADHD, depression, anxiety High interventions 	 Low ID High ADHD, depression, anxiety Latest age of diagnosis High interventions 	 High ID High language delay Earliest age of diagnosis
Rare de novo	** Low evolutionary constraint genes	*** High evolutionary constraint genes	*	**
Rare inherited				***
PGS		Ed. attainment, IQ (depleted) ADHD, depression (enriched)	Ed. attainment (depleted) ADHD, depression (enriched)	
Gene sets enriched	ASD risk genes (enriched)	FMRP target genes (enriched); other autism gene sets (enriched)		Brain-expressed genes (enriched); Satterstrom genes (enriched)
Pathways	Histone and chromatin modification; hormonal pathway	Positive regulation of synapse maturation	DNA repair, cell growth, microtubule activity, calcium channel activity	Membrane depolarization, action potential, sodium channel activity
Developmental timing of genes	Mixed	Mixed	Later expressed genes (Up/Trans-up trends)	Early expressed genes (Down/Trans-down trends)

Table R2.1: Basic summary of phenotypic and genetic findings for each class.

2. The phenotypic clusters are interesting, though it should be noted they likely reflect many potential points of origin, particularly rater effects. Autistic individuals with limited speech and/or global developmental delays/intellectual disability (ID) have very different phenotypic profiles as rated, as one would expect. They are less likely to be accurately assessed for anxiety and mood symptoms, and quite naturally have different average profiles in terms of social behavior. The extent to which the four classes that emerged track with global developmental delay is not adequately considered or discussed in the manuscript. All of the measures employed in the class identification activity have established associations to ID/IQ, and the classes here largely recapitulate expected relationships between ID/IQ and behavioral phenotypes in autism.

We appreciate and understand the reviewer's concerns about rater effects and confounding influence of ID and IQ. However, the fact that our model is strongly reproduced in the SSC cohort suggests that our results are not dependent on rater effects. SSC data was gathered by a small number of trained clinicians through partner institutions, and is therefore not susceptible to the same concerns.

Furthermore, ID/IQ does not drive our clustering. Two of our classes with relatively high and very similar proportions of children with an ID diagnosis (Broadly Impacted with 37.0% ID diagnosis, Mixed ASD with DD with 36.1%, not significantly different p-value of 0.337, FDR = 0.434; Supplementary Table 3) show many significant differences both phenotypically and genetically. Similarly, the two groups with lower levels of ID, Moderate Challenges and Social/Behavioral, also display clearly different phenotypes and genetic patterns. ID diagnosis is quantified here using the measure in the Basic Medical Screening for “Intellectual disability, cognitive impairment, global developmental delay, or borderline intellectual functioning”.

To the reviewer’s point about developmental delay - it is in fact an important aspect of our manuscript - it is one of seven phenotype categories used to define and characterize the classes (Figure 1B, 1C).

To further emphasize these points in the manuscript, we now added the following text:

Discussion: **The current reliance on self-reported data, however, may introduce rater effects, and intellectual disability is a known confounder in assessing social behaviors. Despite these challenges, we show that phenotypic presentation does not reflect a spectrum of intellectual disability, and the self-reported data is both internally consistent with medical history and reproducible in an external cohort with clinician-reported data.**

...

Some of these trends are consistent with known associations between intellectual disability and genetic pathways, though these classes, constructed based on a much richer set of features, provide a more nuanced understanding of autistic individuals and the range of challenges they may face, along with differences in complex genetic architectures that may underlie their condition.

3.

In relation to the reviewer’s comment below, we want to thank you for the time and thought you put into this comment. We address every point you raised in detail below, but first want to emphasize that the power of our analysis comes from the holistic, person-centered approach that considers phenotypic traits and their interactions within individuals, which then allows us to identify complex genetic patterns associated with these co-occurring phenotypes. As expected, our analysis aligns with prior observations from existing literature (e.g. presence of deleterious *de novo* variants), but importantly goes far beyond prior work in not only identifying novel cohesive sub-classes of autism, but also classifying their significant differences along genetic, molecular, developmental, phenotypic, and clinical axes.

Also relevant to all of the reviewer's points below (see detailed answers throughout), our classes do not reflect a continuum of ID, as the spectrum of ASD is not the spectrum of ID, but rather capture the complexity of autism-related genetic, phenotypic, and clinical factors. We have additionally emphasized this throughout the manuscript.

Given the clear role of ID/IQ in formation of the classes, the genetic differences observed are largely expected. The class in which the largest fraction of individuals meet criteria for intellectual disability should have the highest rate of strong acting *de novo* events, as observed here.

Our classes and the associated genetic differences are not explained by ID/IQ. Two of the classes (Broadly Impacted and Mixed ASD with DD) have elevated but statistically indistinguishable proportions of individuals with ID (Supplementary Table 3). While these classes do have more *de novo* variants than non-autistic siblings (all four classes are significantly enriched in deleterious *de novo* variants), we further identify significant differences between the two classes, clearly supporting that they are not driven simply by ID proportions (Figure 3A-C). For example, the Broadly Impacted class has a significantly higher rate of strong acting *de novo* variants than in the Mixed ASD with DD class (Figure 3B left), while only the Mixed ASD with DD class is significantly enriched in deleterious inherited variants (Figure 3B right). Additionally, the class with the lowest rate of strong acting *de novo* variants, Social/Behavioral (Figure 3B), is not the class with the lowest proportion of ID (Figure 2A). Classification by ID/IQ alone would not enable identification of these genetic differences.

Further, observed mutations in that group are expected to reside with genes expressed earliest in development. A suite of papers over the last several years (e.g. Fu et al. 2023) have shown average difference in expression timing between genes associated with ID, autism and schizophrenia. Within autism-associated genes, those also associated to ID are expressed earlier than those not.

Again, our work describes important signals not previously described in prior literature and not explained by the ID proportions. For example, the strongest signal in the developmental gene trajectory analysis for the Broadly Impacted class, the class with the highest ID burden, is observed in genes with the "Trans Up" pattern, which are most highly expressed postnatally (Figure 5B). Furthermore, the Social/Behavioral class is the only class primarily enriched for variation in genes with the "Up" and "Trans Up" patterns (Figure 5B), but it is not the class with the lowest proportion of ID (Figure 2A). Additionally, the Moderate Challenges class, with the lowest proportion of ID, displays a significant burden of deleterious variation in early-expressed genes in multiple cell types (Figure 5B).

Genetic variants most disruptive to human brain development are found in genes expressed earliest, and those associated to ID are more likely to fall within the FMRP pathway. We therefore expect the class of individuals with the greatest ID concentration to show those

genetic patterns [note: given the ID/IQ data available in SPARK, greatest burden of global developmental delay is best indicated by milestone delay data.]

The FMRP pathway is the perfect example of the distinctions our classes provide outside of ID proportions. Among the two classes with statistically indistinguishable proportions of ID, only one class (Broadly Impacted) is significantly enriched in variant impact on the FMRP pathway compared to probands in other classes (Figure R1.3, new Supplementary Figure 11). The other three classes have statistically indistinguishable levels of FMRP enrichment (Figure 4B), even though they have significant differences in ID proportion (Supplementary Table 3).

Additionally, we expect autistic individuals with more severe ID to have a mixture of challenges – as rated – because the presence of ID and autism makes many behavioral phenotypes difficult to assess. On the other hand, we expect individuals in the class with the lowest ID concentration (highest average IQ) to have the lowest rate of autism/ID-associated disruptive variation, and we expect the genes in which the variants they do have are found to be expressed later. We expect the highest IQ group to have the highest rate of anxiety, along with challenges more common in autistic individuals with higher IQ.

Again, our classes capture deeper complexity than the known associations the reviewer describes. Rates of anxiety, depression, and ADHD are highest in the Social/Behavioral class, which does not have the lowest proportion of individuals with ID (Figure 2A). The Moderate Challenges class, with the lowest proportion of ID, has significantly lower rates of anxiety, depression, and ADHD (Supplementary Table 3).

We expect the groups in between to have a mixed genetic architecture, including an elevated rate of rare inherited, putatively damaging variation. The same expectation logic extends to the observed polygenic score differences, with a note that PGS provide clear opportunity to compare variation within versus between the classes, and that analysis would be of benefit.

Thank you for the suggestion. We have now included comparisons between all classes, showing significant differences in common variant signals between them, in addition to previously reported comparisons to non-autistic siblings for every PGS/GWAS (Figure 3A, Supplementary Table 6).

4. It is statistically challenging to compare pathway enrichment, or rare variant rate enrichment, particularly at this sample size. Perhaps I missed it, but it seemed the authors identified enrichment within each group, as compared to null (control expectation). They did not statistically compare enrichment between the groups? This would require multiple testing correction, and apologies if I missed this somewhere.

We have now run statistical testing for all differences across groups, and reported FDR values in the text, as well as marked these relationships in the figures (Figures 1C, 1D,

2B, 3A-C). In order to maintain visual clarity, many of these have been added to the supplement rather than shown in figures (Supplementary Tables 2, 3, 4, 6, 7, 8, and 12). Please refer to reviewer comment 1.2 for the complete description. Please note that FDR values are multiple hypothesis corrected by design, and all statistical comparisons in the manuscript are multiple hypothesis corrected.

Reviewer #3:

Remarks to the Author:

Paper Summary:

Litman, Sauerwald and colleagues took aim at phenotypic heterogeneity within autism. They used an unsupervised, person-centered clustering approach in the SPARK cohort (N of ~5,000) to identify four groups of patients with intuitive interpretability – moderate challenges, broad impact, social/behavioral, and mixed ASD/DD. They suggest these clusters to be ‘significant’ on the basis of goodness of fit metrics, interpretability, and external validity via projection into a smaller SSC cohort (N of ~900). From there, they investigate the genetic architecture of these clusters using a set of polygenic scores and burden of rare de novo and inherited variation. The rare variant burdens were further interpreted for pathway and gene expression signatures. The paper is interesting, creative, pursuing an important question, and well written – clearly, concisely explained and motivated. The methods are clearly explained and visualizations both clean and interpretable. However, despite my enthusiasm for the approach, I have some concerns as to the strength of the evidence supporting the claim of ‘distinct genetic programs.’ These center around wanting a bit more demonstration to the stability of the clusters and how the tests used to identify genetic differences among clusters were performed.

We appreciate the reviewer’s perspective on our work, and the careful consideration they have given to its improvement. We fully address the reviewer’s points below.

Major Comments:

How stable and robustly separable are the clusters?

I understand focusing downstream analysis on a best solution, but I would like more intuition for how stable and separable these particular clusters are. There is emerging evidence (as cited by the authors) that aspects of clinical presentation relate to aspects of genetics. As such, would any clustering solution produce genetic differences that could be justified post hoc? I think (and hope) not. I also like the ‘person-centered’ approach - interesting and feels like a ‘right’ philosophical approach. However, I’d like a bit more demonstration that these particular clusters are ‘special’ / represent some ‘natural kind’ / don’t just trim axes of quantitative covariation.

We thank the reviewer for their suggestions about demonstrating the model's stability and robustness. Based on these suggestions, we have now added several analyses which further support the overall robustness, stability, and separability of the classes (new Supplementary Figure 3b, new Supplementary Figure 5, Figure R3.1).

First, we demonstrate that the 4-class solution presented in the manuscript is highly stable and robust to many different random model initializations (new Supplementary Figure 3b). We trained 2,000 independent models with random initializations, retained the top 100 models (ranked by log likelihood), and observed very strong stability across the randomly initialized models, with averages of 90-97% of individuals assigned to the same group as the main model presented in the manuscript. Additionally, the feature concordances between randomly initialized models and the main model are highly stable across all seven categories of phenotypes.

Supplementary Figure 3b: The 4-class GFMM is highly stable and robust to random model initializations. The heatmap (left) depicts the proportion of individuals assigned to the same group in the randomly initialized models (y-axis) as the main model (x-axis). The barplot (right) quantifies correlations (x-axis) between the feature enrichments of the randomly initialized models and the main model. All proportions represent means and intervals depict the 95% confidence interval.

Second, we have also performed analyses to compare phenotypic variation between classes to phenotypic variation within classes (new Supplementary Figure 5). We found that within-class feature variation is significantly and substantially smaller than between-class feature variation ($p = 9.97e-176$, Supplementary Fig. 5, between-class distribution vs. within-class distribution), further supporting the coherence within and distinction between the classes.

Supplementary Figure 5: Comparison of feature variation between versus within classes.

Next, we addressed the reviewer's concerns about the uniqueness of our clustering solution in uncovering genetic differences (Figure R3.1). Unlike the main clustering solution (Figures 3A-C), we see no statistically significant genetic differences between classes after randomly shuffling the class labels of individuals ($0.375 < \text{FDR} < 0.432$ for *de novo* variants, $0.367 < \text{FDR} < 0.541$ for rare inherited variants, $0.446 < \text{FDR} < 0.899$ for PGS across all possible comparisons, each computed as the minimum of both one-sided tests). This demonstrates that, as the reviewer expected, random clusterings do not result in genetic differences that could be explained or justified post hoc.

Random clustering experiment: shuffled labels

Figure R3.1: Random permutation of cluster labels yields classes with no significant class-class differences.

The goodness of fit metrics use an ‘elbow rule’ heuristic to select four clusters, but the separation of those indices seems more ‘quantitative’ than ‘qualitative.’ It appears (correct me if I am wrong) that all of the metrics for goodness of fit were based on an analysis of the full sample. Does a resampling or subsampling based approach suggest the four clusters are stable to small perturbations of the data? Are subjects frequently flipping classes depending on the context? What is the feature concordance among re- or sub-samples, as tested with the replication? How stable is the feature enrichment to re- or sub- sampling?

We appreciate the reviewer’s suggestion to test the stability of the model to robustness tests such as subsampling or resampling. Based on these suggestions, we have performed analyses which demonstrate that the model is highly robust to perturbations of the data, even at a subsampling rate of 50% (new Supplementary Figure 3a). We trained 100 independent models (each with 20 random initializations) with a subsampling rate of 50% of individuals in the sample, and found very strong stability across the subsampled models, with averages of 88-96% of individuals assigned to the same group as the full model. Additionally, the feature concordances between subsampled models and the main model are highly stable across all seven categories of phenotypes.

Supplementary Figure 3a: The model is highly stable to subsampling perturbations of the data (50% subsampling rate). The heatmap (left) represents the proportion of individuals assigned to the same group in the subsampled models (y-axis) as the full model (x-axis). The barplot (right) quantifies correlations (x-axis) between the feature enrichments of the subsampled models and the full model. All proportions represent means and all intervals depict 95% CI.

Interpretability was used as one metric to choose four clusters, but the metrics for comparator solutions are not in the paper. Four clusters had the ‘best phenotypic separation and most clinically coherent classes’ but it would be useful to see that data. Could the authors provide, e.g., metrics as in Figure 1 for a few candidate models – 3-6 clusters?

We thank the reviewer for their suggestion. We have now provided plots similar to those shown in Figure 1 for 3-, 5-, and 6-class models (new Supplementary Figure 2). We demonstrate that these class solutions all provide either too general or less interpretable clusters as compared to the 4-class solution.

The 3-class model identified 3 of 4 classes in the 4-class model, with the exception of the Broadly Impacted class, which, based on our analyses, emerges as phenotypically and genetically distinct in the 4-class solution. The 5-class solution largely preserves two of the same classes from the 4-class model, while splitting the other two groups into three groups with challenging interpretations due to small differences, large overlaps, and mixed enrichment patterns in the phenotype categories. Finally, the 6-class solution presents an even less interpretable model, with four classes that are too fine-grained and have non-intuitive enrichment patterns. Additionally, the majority of classes in the 5- and 6-class models have under 1,000 individuals, making statistical testing in downstream analyses very challenging. At 6 classes, the model may also be overfitting to our cohort, capturing patterns which may not be generalizable across the wider population. With larger sample sizes, it is possible that a more complex solution will

prove to be optimal, but the 4-class solution is the best balance of descriptive and interpretable with our cohort.

Supplementary Figure 2: Model characteristics for 3-, 5-, and 6-class models.

How does the GFMM handle potential confounding covariates? It is mentioned “1-step StepMix with age and sex as covariates” – how are these treated by the clustering? What about genetic ancestry? It is a little unclear if a multi- or single- ancestry sample was used for the clustering. If multi-ancestry, this can be problematic as associated social factors could impact clustering. Can PCs be adjusted or ancestry tested in other ways for impact on clustering?

The model directly accounts for covariates by conditioning the class distributions on each covariate. Intuitively, since our model includes age and sex as covariates, the mixture model handles this by estimating the probability of each individual belonging to a certain latent class conditioned on their sex and age. The model takes into account that, statistically, males and females or individuals of different ages might have different

baseline distributions in the phenotype features. This means it will expect some phenotypic differences due to these covariates and adjust the class probabilities accordingly.

Next, we demonstrate that self-reported race does not influence the clustering solution (Figure R3.2). We expect that self-reported race (available for the majority of the sample) would have a greater impact on social factors related to phenotype measures than genetic ancestry, so we tested whether including race as a covariate has a significant impact on the model predictions (this also aligns with our overall approach where clustering is based on phenotypes, not genetics). We retrained the model on this sample and found nearly perfect concordance between models trained with and without race as a covariate, with a minimum overlap proportion of 0.998 across classes (Figure R3.2). Thus, we do not see any differences in the model predictions when controlling for race.

Figure R3.2: Overlap of individuals between two clustering solutions with different covariate sets.

For the SSC replication, is shuffling the test sample labels the appropriate null? Would it not make more sense to shuffle the labels of the SPARK, project those clusters into the SSC, and test concordance? i.e., is the concordance between SPARK and SSC of the real cluster projection better than a permuted cluster projection? This would better support stability of those particular clusters, no?

We thank the reviewer for their insightful suggestion. Based on this suggestion, we have now added the reviewer-proposed replication analysis, which further supports the robustness of the replication and stability of the clusters (added to Supplementary Figure 9). We trained 1,000 independent models on shuffled SPARK phenotype labels,

projected those permuted clusters onto the SSC, and tested the concordance with the true SPARK labels. We found that all correlations computed with the permuted cluster projections fail to surpass the true correlation from the real cluster projection ($r = 0.926$, $p < 1e-4$ for comparison between true correlation and permuted cluster projection correlations).

Supplementary Figure 9: Permuted cluster projections (blue) validate the true cluster projection correlation between the SSC and SPARK ($r = 0.926$, red line).

How much additional support do the BMS and parent/self-report give? Is their enrichment not tautological? Many of the outcomes used to confirm the clusters are 'naturally' related to the instruments used to construct the clusters (CBCL -> BMS; BhX -> parent/self-report milestones), so are we validating the clusters or the construct validity of instruments (i.e., 'validating' height in inches with height in centimeters)? I am not sure how to address this, but it is worth discussing/tempering this additional evidence that the clusters are 'phenotypically distinct.' This issue aside, the enrichments are reported/tested against the non-autistic siblings, not in a competitive way against the classes, so we cannot know if the differences in enrichment among groups (the key concept) is significant, as it is not reported.

Thank you for the suggestion. We have now computed statistics comparing the classes directly for the data shown in Figures 1C, 1D, 2A, and 2B, and reported these values in full in Supplementary Tables 2, 3, and 4, showing many significant phenotypic differences between classes. We also note that Figure 1B already reflects differences among probands, as feature enrichments for each phenotype category were computed within the pool of probands, with each class compared to out-of-group probands. Additionally, we refer to the variances within- and between- classes above (new

Supplementary Figure 5) to further demonstrate the distinct differences in phenotypes across groups.

To address the BMS-based analysis, we appreciate the reviewer's point. That said, the BMS was the best available source of data outside of our training set. We have added the following sentences to the text to temper the discussion of this analysis (new text in red):

Results: *These diagnosis data represent the best available external validation, though the natural associations between the behavioral questionnaires that our model was trained on and behavioral diagnoses result in this being a not fully orthogonal validation set. The consistency observed here, however, does further support the validity of the self-reported data.* Together, these analyses of medical features show that the four classes identified by the GFMM are phenotypically *consistent*, supporting their separation in genetic analyses.

Evidence for genetic differences

As I understand, the significance of (all of) the genetic enrichments stems from differences in dichotomized significance of case-control tests, i.e., sub-group A is significantly different from controls, but sub-group B is not. These kinds of inferences are problematic and can represent false dichotomies/winner's curse biases. A formal test of the difference in the case-control enrichments or difference between the case sub-groups, directly, is (at minimum) needed to justify the claims of distinct genetic programs. A pessimistic read suggests nearly all of the subgroup genetic enrichments are not significant between case-groups, given the nearly overlapping standard errors (where provided). I believe this affects results from Figures 3A-C, 4A-C, and 5B.

We thank the reviewer for the suggestion; we have now run statistical testing for all differences across groups, and reported FDR values in the text, as well as marked many of these relationships in the figures (Figures 1C, 1D, 2B, 3A-C). In order to maintain visual clarity, many of these have been added to the supplement rather than shown in figures. We have also added Supplementary Figure 11 to portray gene set comparisons within the pool of probands. The statistics were additionally computed for all comparisons in Figures 2A, 4A, and 5B. We report the full results of these comparisons in updated and new Supplementary Tables 2, 3, 4, 6, 7, 8, and 12. Notably, the analysis in Figure 4C does not include any comparisons to siblings as a control group; this is a gene-set based analysis rather than person-based enrichment, and therefore does not lend itself to statistical testing between case-groups. Please note that all relevant figures and tables were also updated with the latest whole exome release from SPARK (n=414 new exomes for our cohort).

Rather than contrasting significance levels (which can be driven by differences in the size of the groups), it would be more useful to contrast effect sizes, and to use 95% CI instead of SEM (easier to intuit subgroup differences as meaningful). When an enrichment is proposed for one sub-group, it would be useful to show its level and 95% for all other subgroups (not done in panel 4C).

We thank the reviewer for their suggestions. With the addition of direct comparisons between classes, versus the original focus on contrasting the significance levels (vs. siblings), we now have a direct assessment of how classes compare (see Figures 3A-C). Additionally, we have computed Cohen's *d* for all comparisons of continuous values, now reported in Supplementary Tables 2, 4, and 6, and fold change for all comparisons of binary or categorical phenotypic variables and rare variant enrichments, which are reported in Supplementary Tables 3, 4, 7, 8, and 12. Also, based on the reviewer's suggestion, we have updated all relevant figures with the 95% CI, in place of the SEM.

Some of these "enrichment tests" (4C, 5B) use complicated heuristic statistics and may not have intuitive effect sizes with easy estimates of variance, and resampling procedures could be complicated. I am less worried about these more descriptive analysis (although see my comment about 4C above), but for 3A-C and 4A-B, in particular, I think the sub-group differences should be more rigorously tested.

This is a very valid point, and, as mentioned above, we have now run statistical testing for all differences across groups, and reported FDR values in the text, as well as marked many of these relationships in the figures (Figures 1C, 1D, 2B, 3A-C). The statistics were additionally computed for comparisons in Figures 2A, 4A, and 5B. We report the full results of these comparisons in updated and new Supplementary Tables 2, 3, 4, 6, 7, 8, and 12.

Notably, the analysis in Figure 4C does not include any comparisons to siblings as a control group; this is a gene-set based analysis rather than person-based enrichment, and therefore does not lend itself to statistical testing between case-groups. This is a standard approach to pathway enrichment analysis used extensively in the literature.

Minor Comments:

For genetic analysis, it mentions self-report European, but this can/should be confirmed with genetic ancestry PCs.

We appreciate the reviewer's suggestion to confirm self-reported ancestry with PCA. We demonstrate below that genetic ancestry in PC space is consistent with the self-reported data (new Supplementary Figure 10).

Supplementary Figure 10: Ancestry PCA plot for genotyped cohort. Colored by self-reported race.

ASD has increased in prevalence and breadth over recent decades. Some previous subgroups are gone (Asperger's) and some individuals previous categorized as intellectual disability or developmental delay may now fall in ASD. Could the authors discuss/speculate if their clusters are 'rediscovering' previous clinical groups? Is the move to a more 'inclusive ASD' perhaps not supported by etiological data?

We appreciate this interesting question! The clinicians we worked with propose that the inclusivity of the modern ASD diagnosis is more pertinent to our uncertainty about the exact subtyping rather than a belief that ASD is in fact a single homogenous condition. Our clusters support the heterogeneity of ASD and the existence of subtypes, though we do not address whether these subtypes should be separated into multiple conditions with differing diagnostic criteria. There seems to be consensus among clinicians that there are truly multiple types of ASD in some sense, though not enough consensus on the details of what those types are to formally separate them at this time. Our work aims to advance this understanding of subgroups, and perhaps additional data-driven analyses can continue to build consensus and insight in this direction. Though we are hesitant to speculate too much on the correlation between our classes and prior clinical groups without any supporting data, the question is very interesting to think about!

Is this use of non-autistic siblings as a control group the most efficient? From my read, non-autistic siblings, as a pool, are used to estimate some aspects of enrichment (Figure 3A), and not via a paired analysis. I wonder if this doesn't mix apples with oranges. Take sibships A1-A2 and B1-B2 (A1/B1 case, and A2/B2 control). The differences in PGS between A1-A2 and B1-B2 represent within family genetic variance (segregation variance, roughly half the total

genetic variance) while the difference between A1-B2 and B1-A2 represent both within and between family genetic variance (the full genetic variance. The genetic distance (e.g., mean PGS difference) between cases and controls is then a mixture of within and between family differences, with within family differences expected to be smaller). In other words, we expect the PGS of the ASD sibs to be closer to their non-ASD sib than another cases non-ASD sib. Could this affect the PGS (or rare variant?) enrichments? Differences between cases and controls for larger subgroups will be more weighted towards within family differences, while smaller groups to between family differences, as larger subgroups will have more sibs in the controls, smaller subgroups, fewer sibs. I wonder if this mixing of within and between family contrasts doesn't add noise to the enrichment tests. Case-subgroup contrasts would not suffer this – A1-B1 differences capture differences in the full genetic variance (within and between family).

We appreciate the reviewer's important consideration of the implications of comparing genetics using siblings as a control group. The inclusion of comparisons directly between case-subgroups, now seen in Figures 1C, 1D, 2B, 3A-C and Supplementary Figure 11, and added in Supplementary Tables 2, 3, 4, 6, 7, 8, and 12, should alleviate these concerns by removing the paired ancestries between family members, as there are relatively few siblings among the cohort of probands.

Is parental history of disorders available? This could be an independent replication of genetic differences if the clusters had parents enriched for the same disorders as they have enriched PGS. It turns out these can be relatively independent measure of genetic liability (e.g., PMID: 36347255).

We thank the reviewer for this extremely interesting suggestion. Unfortunately, SPARK does not have sufficient parent diagnostic data to run the recommended analysis.

References

1. Masyn, K. E. Latent Class Analysis and Finite Mixture Modeling. in *The Oxford Handbook of Quantitative Methods* (ed. Little, T. D.) (2013).
2. Nylund-Gibson, K. & Choi, A. Y. Ten frequently asked questions about latent class analysis. *Translational Issues in Psychological Science* **4**, 440–461 (2018).
3. Sinha, P., Calfee, C. S. & Delucchi, K. L. Practitioner's Guide to Latent Class Analysis: Methodological Considerations and Common Pitfalls. *Crit. Care Med.* E63–E79 (2020).
4. Garber, K. B., Visootsak, J. & Warren, S. T. Fragile X syndrome. *Eur. J. Hum. Genet.* **16**, 666–672 (2008).

5. Morin, S. *et al.* StepMix: A Python Package for Pseudo-Likelihood Estimation of Generalized Mixture Models with External Variables. *arXiv [stat.ME]* (2023).

Reviewer #1:

Remarks to the Author:

The authors have done a great job in addressing many of my comments. This is an important concept to address and interrogate. Upon re-reading the reviews and the paper, however, I do agree with reviewer 2's first point (which is the spirit behind my first point about the number of classes). I do think that distinctive is not the right word over here for a few reasons. First, if I were to look at the elbow plots and the fit statistics (I still don't see an elbow, I see a gentle curve), I would pick out a two-class solution as the best one. This may likely reflect a separation based on ID/DDD. It would still make sense clinically. I don't think the authors are incorrect, and I'm merely highlighting the subjectivity of this process. Second, although there is greater variance within than between, there isn't a clear genetic or phenotypic separation between the groups to use the term distinctive. The interested but non-specialist reader would walk away with the message that there are four clear, separate groups among autistic individuals. I don't think this is what the manuscript is saying (and I don't think this is what the authors intend to suggest). Rather, I think the manuscript points to certain dimensions in a multivariate phenotypic space that can be used to cluster autistic individuals into groups. These groups differ in their genetic profile. A litmus test would be to ask if you see an autistic child with a combination of phenotypes can you easily classify them into one of four groups? I suspect not.

So, I would strongly emphasise against the use of the word distinctive (how about "differing" or "dissimilar" instead?). I'm also not sure about the word coherent - what do you mean by coherent here? Do you mean that it comes together nicely? Apologies, I don't seem to get it (and I'm not sure if it's needed here).

So, in sum, I think the authors have done a great job addressing my comments. I think this is an important contribution in the ongoing debate on the optimal way to characterise autism. That said, in the spirit of the debate, I would recommend using a different word from distinct and not using the term coherent as these terms belie the subjective and fuzzy nature of these groupings.

We greatly appreciate the reviewer's comments and perspective on this work. To address the concerns about word choice, we have changed the title to "Decomposition of phenotypic heterogeneity in autism reveals underlying genetic programs", and replaced all instances of the words "distinct" and "coherent" that referred to the four groups. They have been replaced with the suggested words to clarify the messaging and reduce subjectivity, hopefully alleviating the concerns of both reviewers.